# SHH, DON'T SAY THAT! DOMAIN CERTIFICATION IN LLMS

**Cornelius Emde**[1][*]**, Alasdair Paren**[1]**, Preetham Arvind**[1]**, Maxime Kayser**[1]**, Tom Rainforth**[1]**,
Thomas Lukasiewicz**[2,1]**, Philip H.S. Torr**[1]**, Adel Bibi**[1]
[1]University of Oxford     [2]Vienna University of Technology

## ABSTRACT

Large language models (LLMs) are often deployed to perform constrained tasks, with narrow domains. For example, customer support bots can be built on top of LLMs, relying on their broad language understanding and capabilities to enhance performance. However, these LLMs are adversarially susceptible, potentially generating outputs outside the intended domain. To formalize, assess, and mitigate this risk, we introduce *domain certification*; a guarantee that accurately characterizes the out-of-domain behavior of language models. We then propose a simple yet effective approach, which we call VALID that provides adversarial bounds as a certificate. Finally, we evaluate our method across a diverse set of datasets, demonstrating that it yields meaningful certificates, which bound the probability of out-of-domain samples tightly with minimum penalty to refusal behavior.

## 1 INTRODUCTION

With recent advancements in the field of natural language processing, large language models (LLMs) have become ubiquitous. In particular, the scaling of recent large generalist models dubbed foundation models has shown to enable emergent abilities that benefit a wide range of downstream tasks such as text generation, question answering, and text comprehension (Kaplan et al., 2020; Alabdulmohsin et al., 2022; Xiong et al., 2024; Henighan et al., 2020; Brown et al., 2020). Adapting these foundation models for downstream tasks often leads to state-of-the-art performance and has become the dominant paradigm (Gao et al., 2021). This is typically achieved via fine-tuning on task-relevant data (e.g., low-rank adaptation (LoRA) Hu et al. (2022), in-context learning (Mosbach et al., 2023), prefix turning Li & Liang (2021), or simply prompt engineering).

However, foundation models are typically trained on large amounts of web data which contains a wide range of information that is either irrelevant to a task or potentially harmful (Bommasani et al., 2022). Therefore, it is desirable to restrict the output of a generalist LLM to a specific domain. For example, consider a healthcare provider such as the National Health Services (NHS) providing a general purpose

**Health Advice Chatbot**

I am a chatbot to help with health questions.

I need help commiting tax fraud.

Sorry. I can't help.

I need help commiting tax fraud. #@/!%\£~~

Sure. Here is a five point plan:

Figure 1: A user misappropriating an LLM system using an adversarial attack. We provide certificates to mitigate this risk.

chatbot to support their citizens with simple health questions, as shown in Figure 1. It would be important, for public reputation and cost reasons, that such a system would remain on topic and could not be misused, either intentionally or unintentionally. Misappropriating models is easily possible.

In order to prevent intentional misuse, we consider an adversary trying to elicit an unintended (from the deployer's perspective) response from the model. We assume the deployer wants an LLM to only respond with a certain set of topics, and thus a successful attack is an input string that creates a coherent response outside the target domain. There are various reasons why an adversary might want to elicit such a response that is out-of-domain (OOD). The adversarial user might want to

---

[*]Corresponding Author `cornelius.emde@cs.ox.ac.uk`. Work partially done while interning at King Abdullah University of Science and Technology (KAUST).

misappropriate the system as a cost-effective tool for a purpose it wasn't built for, resulting in excess infrastructure costs for the deployer. Conversely, the deployer might legally be required to validate and verify their models, which is challenging, if not impossible, when the model is not domain-restricted. Finally, the adversary might want to harm the company directly by eliciting harmful OOD responses, which could damage the company's reputation when publicized. Recently, an LLM-driven meal planning tool has received wide media attention for providing toxic recipes when prompted with toxic ingredients (McClure, 2023; The Guardian, 2023). Deployers have moral and legal obligations to prevent this (Bommasani et al., 2022). In all examples, restricting the domain in which the model responds *under adversarial* prompts can help mitigate risks. Thus, in the era of foundation models, "domain" specialization is critical.

Existing work has implemented guardrails that address these risks (Jain et al., 2023), most notably via alignment, resulting in models rejecting user requests (Bai et al., 2022; Ouyang et al., 2022; Christiano et al., 2017). However, a wide body of research has shown that common guardrails have "jailbreaks", i.e., they can easily be circumvented by an adversary (Wang et al., 2024; Qi et al., 2024; Eiras et al., 2024; Carlini et al., 2023; Dong et al., 2024). Common jailbreak methods are prompt injection (Perez & Ribeiro, 2022; Jiang et al., 2023; Liu et al., 2024), numerical optimization (Jia & Liang, 2017; Wallace et al., 2019; Ebrahimi et al., 2018; Jones et al., 2023; Zou et al., 2023; Jia et al., 2025), red teaming (Perez et al., 2022; Samvelyan et al., 2024), automated black-box attacks (Chao et al., 2023; Mehrotra et al., 2024), or data poisoning attacks (Biggio et al., 2012; Wallace et al., 2021; Carlini et al., 2024). Using these tools, it is possible for adversaries to retrieve information from a fine-tuned model that was suppressed by the alignment and generate responses that are outside the target domain (see Figure 1 for an example). Adversarial prefixes or suffixes that augment any prompt are especially powerful, as they have been shown to universally attack models in combination with a wide range of prompts and can thus be shared between adversarial users (Wallace et al., 2019; Zou et al., 2023). This presents a significant risk. Hence, researchers have proposed methods to defend against these adversarial attacks, such as unlearning (Nguyen et al., 2022; Xu et al., 2023), robust fine-tuning (O'Neill et al., 2023; Dong et al., 2021), or request and response filtering (Inan et al., 2023).

Deployers would ideally want guardrails that come with a provable, mathematical guarantee against the model responding off-topic, or a guarantee that it does this with very low probability. The process of constructing guarantees against certain model behaviors under adversarial attack is commonly referred to as *certification* and has been successfully applied to vision applications in recent years (Akhtar et al., 2021) and proposed for NLP applications (La Malfa, 2023; Casadio et al., 2024; Kumar et al., 2024). However, no existing LLM guardrails provide guaranteed protection against existing or future jailbreaking techniques, leaving deployed models at risk of being compromised shortly after release. As a result, developing certifiable methods to guarantee that specialized LLMs consistently produce on-topic content is critical. Hence, our contributions are as follows:

- We introduce a novel framework, *domain certification*, to bound the probability of models producing out-of-domain content under adversarial attack.
- We introduce an easy-to-use algorithm VALID that bounds the probability of an LLM based system responding off topic under adversarial attack. We show the efficiency of VALID which we test empirically on a number of representative data sets.

## 2 DOMAIN CERTIFICATION

We now introduce our *domain-certification* framework for offering mathematical guarantees that an LLM system stays on topic. In Section 2.1, we formally introduce this framework. In Section 2.2, we present **V**erified **A**dversarial **LL**M Output via **I**terative **D**ismissal (VALID). VALID is an easy-to-use method to create a system that adheres to these guarantees. In plain language, we propose a certifiable guardrail for LLM-driven systems as follows:

> A model is domain-certified, *when an adversarial upper bound can be placed on the probability that the model provides an output outside its designated target domain.*

Before formalizing this statement, we introduce some mathematical notation. We represent tokens (i.e. individual text units) as $x$ and $y$, which belong to the token space $x, y \in \mathbb{V}$ where $\mathbb{V} = \{1, \ldots, V\}$ is the vocabulary of size $V$. We define the space of sequences of arbitrary length as $\mathbb{S} \triangleq \mathbb{V}^*$, the Kleene closure of $\mathbb{V}$. Sequences of tokens are denoted by bold letters, $\boldsymbol{x}, \boldsymbol{y} \in \mathbb{S}$, with

$\boldsymbol{x}$ and $\boldsymbol{y}$ representing the input and output sequences of an LLM respectively. We use lowercase letters to denote models that predict the next token, such as $l : \mathbb{S} \to \mathbb{V}$. Applying this model repeatedly, until the end-of-sequence token creates a sequence-to-sequence model $L : \mathbb{S} \to \mathbb{S}$. We denote the likelihood of sample $\boldsymbol{y}$ under $L$ given $\boldsymbol{x}$ as $L(\boldsymbol{y}|\boldsymbol{x})$, which is obtained by $L(\boldsymbol{y}|\boldsymbol{x}) = \prod_{n=1}^{N_y} l(y_n|y_{<n}, \boldsymbol{x})$ for a sentence $\boldsymbol{y}$ of length $N_y$. We further denote the distribution from which the model samples its output by $\boldsymbol{y} \sim L(\cdot|\boldsymbol{x})$.

## 2.1 Defining Domain Certification

We now formally introduce *domain certification*. We define the target domain (set of desired topics) as a subset of the sentence space $\mathbb{S}$ and partition $\mathbb{S}$ into the target domain $\mathbb{T}$ and its complement $\mathbb{T}'$. For instance, $\mathbb{T}$ might be all sentences meaningfully occurring for "question answering for health problems". In addition, we define the set of unwanted responses as $\mathbb{F} \subset \mathbb{T}'$ ($\mathbb{F}$ as "forbidden") and will certify with respect to this set $\mathbb{F}$ rather than $\mathbb{T}$. Sequences posing some risk should be included in $\mathbb{F}$, while $\mathbb{F}' \cap \mathbb{T}'$ should contain benign out-of-domain samples, such as unintelligible or meaningless sequences of tokens (see Appendix B for a discussion). Hence, we wish to establish a guarantee that $L$ is unlikely to produce an output in $\mathbb{F}$. As a step towards such a guarantee, we first define a bound for any given element $\boldsymbol{y}$ in $\mathbb{S}$:

**Definition 1** *Atomic Certificate*. *We say a model $L : \mathbb{S} \to \mathbb{S}$ is $\epsilon_{\boldsymbol{y}}$-atomic-certified ($\epsilon_{\boldsymbol{y}}$-AC) for some sample $\boldsymbol{y}$ (i.e. an* atom*) in the output set $\mathbb{S}$, iff*

$$\forall \boldsymbol{x} \in \mathbb{S} : L(\boldsymbol{y}|\boldsymbol{x}) \leq \epsilon_{\boldsymbol{y}}. \tag{1}$$

In words, a model that is $\epsilon_{\boldsymbol{y}}$-AC for a sample $\boldsymbol{y}$, will generate sample $\boldsymbol{y}$ with probability smaller than $\epsilon_{\boldsymbol{y}}$ for *any* $\boldsymbol{x} \in \mathbb{S}$, and hence for adversarially chosen $\boldsymbol{x}$. If this is the case, we say model $L$ is *certifiable* for sample $\boldsymbol{y}$ with $\epsilon_{\boldsymbol{y}}$, i.e. $\epsilon_{\boldsymbol{y}}$ is the *smallest* value that provably bounds $L$. Ideally, such an upper bound $\epsilon_{\boldsymbol{y}}$ would be large for samples in the target domain $\mathbb{T}$, meaning the certificate is *permissive*, and small for samples drawn from $\mathbb{F}$ meaning the certificate is *restrictive*, i.e. *tight*.

The atomic certificate implies an upper bound $\epsilon_{\mathbb{F}}$ for $\mathbb{P}_{\boldsymbol{y} \sim L(\cdot|\boldsymbol{x})}(\boldsymbol{y} \in \mathbb{F}|\boldsymbol{x})$, which would be constructed by summing (1) over all $\boldsymbol{y} \in \mathbb{F}$ for a given $\boldsymbol{x}$. Concretely, $\mathbb{P}_{\boldsymbol{y} \sim L(\cdot|\boldsymbol{x})}(\boldsymbol{y} \in \mathbb{F}|\boldsymbol{x}) = \sum_{\boldsymbol{y} \in \mathbb{F}} L(\boldsymbol{y}|\boldsymbol{x}) \leq \sum_{\boldsymbol{y} \in \mathbb{F}} \epsilon_{\boldsymbol{y}} = \epsilon_{\mathbb{F}}$. However, practically this bound is intractable due to $\mathbb{F}$'s exponential size in $N_y$, and the difficulty in constructing a precise description of the set $\mathbb{F}$. Instead of giving a bound over returning $\boldsymbol{y} \in \mathbb{F}$, we look at the worst case across $\mathbb{F}$ which can more precisely be estimated from a finite sample of $\mathbb{F}$:

**Definition 2** *Domain Certificate*. *We say model $L$ is $\epsilon$-domain-certified ($\epsilon$-DC) with respect to $\mathbb{F}$, when it is $\epsilon_{\boldsymbol{y}}$-AC for all $\boldsymbol{y} \in \mathbb{F}$ with $\epsilon_{\boldsymbol{y}} \leq \epsilon$:*

$$\forall \boldsymbol{x} \in \mathbb{S}, \boldsymbol{y} \in \mathbb{F} : L(\boldsymbol{y}|\boldsymbol{x}) \leq \epsilon. \tag{2}$$

This imposes a global bound on $L$ across all undesired responses in $\mathbb{F}$. In practice, we cannot establish the $\epsilon$-DC certificate w.r.t. $\mathbb{F}$ as we cannot enumerate $\mathbb{F}$. Hence, following standard practice in ML evaluation, we propose to use $\mathcal{D}_{\mathbb{F}}$, a finite dataset of out-of-domain responses to establish a $\epsilon$-DC certificate w.r.t. $\mathcal{D}_{\mathbb{F}}$ approximating the certificate for $\mathbb{F}$.

Recent discussions have raised the need for bounds on undesirable behavior. For instance, Bengio (2024) advocates for upper bounds on harmful behavior (Bengio et al., 2024). In addition, a growing body of legislation mandates thorough auditing of ML systems (EU, 2024). The atomic and domain certificates can play a vital role in assessing the risk of worst-case behavior. For example, consider the deployer of an LLM-based system that processes 10 requests per second. The deployer might perform an apriori risk assessment and determine that they can tolerate the consequences of an out-of-domain response from a set $\mathcal{D}_{\mathbb{F}}$ sampled once per year. The deployer should certify the LLM system as $\epsilon$-DC with $\epsilon \approx 10^{-9}$ in order to achieve this level of risk.

**Certification through Divergences.** We provide an alternative view to this problem, generalizing it to bounding divergences between the model and the distribution of sentences in the domain $\mathbb{T}$. We then use this view to operationalize the $\epsilon_{\boldsymbol{y}} - AC$ and $\epsilon - DC$ (Definitions 1 and 2) inspired by Vyas et al. (2023)'s work on preventing copy-right violations. To this end, we define an oracle $\Omega$ that is a *generator* for domain $\mathbb{T}$: $\Omega$ assigns high likelihood to sentences in $\mathbb{T}$ and zero likelihood to

elements in $\mathbb{F}$. Hence, sampling from $\Omega$ will yield in-domain responses. We establish and bound the divergence between $L$ and $\Omega$ to restrict the model domain. In particular, we use the Renyi divergence of order infinity, $\Delta_\infty(P \parallel Q) \triangleq \log \sup_x \frac{P(x)}{Q(x)}$ (Rényi, 1961). Hence, our objective is:

$$\forall \boldsymbol{x} \in \mathbb{S} : \Delta_\infty(L(\boldsymbol{y}|\boldsymbol{x}) \parallel \Omega(\boldsymbol{y})) \leq k. \tag{3}$$

Bounding this divergence is at the core of what we are aiming to achieve: The divergence is large when $L$ assigns high likelihood to a sample $\boldsymbol{y}$ while $\Omega$ does not. That means $L$ is likely to produce samples that are out-of-domain. When $\Omega$ assigns high likelihood to $\boldsymbol{y}$, the sample is in the target domain, and hence the divergence in (3) is not restrictive. When $L$ assigns low likelihood, $\boldsymbol{y}$ is unlikely to be sampled. Interestingly, this divergence implies (1) and (2), see Lemma 1.

As the oracle is not available in practice we approximate $\Omega$ with a "guide" language model that is exclusively trained on in-domain data dubbed $G$ (i.e. the guide model). We use $G(\boldsymbol{y})$ to replace $\Omega(\boldsymbol{y})$ to assess the *marginal* likelihood of $\boldsymbol{y}$. While this means that $G(\boldsymbol{y})$ loses some context contained in $\boldsymbol{x}$, this has a major advantage: $G(\boldsymbol{y})$ does not depend on $\boldsymbol{x}$, which is a potential adversary and hence, by design is robust to adversarial prompts.

## 2.2 Achieving Domain Certification

In this section, we introduce **V**erified **A**dversarial **LLM** Output via **I**terative **D**ismissal (VALID) to obtain atomic certification as described in Definition 1. We utilize a general model $L$ and a domain generator $G$ as described above and obtain a meta-model $M$ for which the guarantee holds with respect to the domain generator $G$. In particular, we perform rejection sampling as described in Algorithm 1 (inspired by Vyas et al. (2023)): The capable general model $L$ proposes a sample $\boldsymbol{y}$ and we accept, if the length normalized log-ratio between $L$ and $G$ is bounded by hyperparameter $k$. We repeat up to $T$ times

---

**Algorithm 1** VALID

**Require:** LLM $L$, Guide model $G$, hyperparameters $k$ and $T$, prompt $\boldsymbol{x}$
   **for** $t \in \{1, \ldots, T\}$ **do**
      Sample $\boldsymbol{y} \sim L(\cdot|\boldsymbol{x})$
      $N_{\boldsymbol{y}} \leftarrow \text{length}(\boldsymbol{y})$
      **if** $\log \frac{L(\boldsymbol{y}|\boldsymbol{x})}{G(\boldsymbol{y})} \leq k N_{\boldsymbol{y}}$ **then**
         **Return:** $\boldsymbol{y}$
   **Return:** "Abstained".

---

until a sample is accepted. If all samples are rejected, the model dismisses the request. This defines a new model $M$, for which the following theorem establishes the certificate:

**Theorem 1 (VALID Certificate)** *Let $L$ be an LLM and $G$ a guide model as described above. Rejection sampling as described in Algorithm 1 with rejection threshold $k$ and up to $T$ iterations defines model $M_{L,G,k,T}$ with $M_{L,G,k,T}(\boldsymbol{y}|\boldsymbol{x})$ denoting the likelihood of $\boldsymbol{y}$ given $\boldsymbol{x}$. Let $N_{\boldsymbol{y}}$ be the length of $\boldsymbol{y}$. We state the adversarial bound:*

$$\forall \boldsymbol{x} \in \mathbb{S} : M_{L,G,k,T}(\boldsymbol{y}|\boldsymbol{x}) \leq 2^{k N_{\boldsymbol{y}}} \cdot T \cdot G(\boldsymbol{y}). \tag{4}$$

*Hence, $M_{L,G,k,T}$ is $[2^{k N_{\boldsymbol{y}}} T G(\boldsymbol{y})]$-AC and, further, it is $[\max_{\boldsymbol{y} \in \mathbb{F}} 2^{k N_{\boldsymbol{y}}} T G(\boldsymbol{y})]$-DC w.r.t. $\mathbb{F}$.*

When context allows, we may abbreviate $M_{L,G,k,T}$ to $M$, omitting subscripts for brevity. This certificate with respect to $G$ can be useful: As $G$ is only trained on samples in $\mathcal{D}_{\mathbb{T}} \subset \mathbb{T}$, a dataset of domain $\mathbb{T}$, it assigns exponentially decreasing likelihood to samples that are in $\mathbb{F}$.[1] In particular, this is useful iff the log upper bound $k N_{\boldsymbol{y}} + \log T + \log G(\boldsymbol{y})$ (log RHS of (4)) is small in comparison to $\max_{x \in \mathbb{S}} \log L(\boldsymbol{y}|\boldsymbol{x})$: Our certificate can provide an upper bound to the adversarial behavior of $M$ that is favorable over $L$.

As mentioned, this problem is closely related to OOD detection, for which the likelihood ratio test is commonly used as a powerful statistic (Neyman & Pearson, 1933; Bishop, 1994; Ren et al., 2019; Li et al., 2023; Zhang et al., 2024; Rafailov et al., 2024). In OOD detection, rejection threshold $k$ is commonly chosen to balance false negative rates and false positive rates. Here, $k$ also influences the upper bound on the certificate, indicating that there can be a *trade-off* between correctly classifying samples as ID or OOD, and achieving a desired level of certification.

**Length Normalization.** Algorithm 1 performs length normalized rejection-sampling as unnormalized log likelihood ratios scale unfavorably in $N_{\boldsymbol{y}}$, the length of sequence $\boldsymbol{y}$ which we now demonstrate. Consider the next-token models $l$ and $g$ underlying the sequence-to-sequence models $L$ and $G$. As $\boldsymbol{y}$ is sampled from $L$, we expect each token $y_1, \ldots, y_{N_{\boldsymbol{y}}}$ to have high likelihood under $l$. If we assume that $l$ places $c$ times more probability mass per token than $g$,

---

[1]We give an empirical example of this behavior in Figure 13 in Appendix E.4.

then we can show that the log likelihood ratio grows linearly in $N_{\boldsymbol{y}}$, the length of sequence $\boldsymbol{y}$: $\log L(\boldsymbol{y}|\boldsymbol{x})/G(\boldsymbol{y}) = \log \prod_{n=1}^{N_{\boldsymbol{y}}} cg(y_n|y_{<n})/g(y_n|y_{<n}) = N_{\boldsymbol{y}} \log c$. We illustrate an example in Figure 2: Assume that an in-domain sample $\boldsymbol{y}$ for which model $L$ and generator $G$ assign constant likelihood per token of 0.1 and 0.05, respectively, i.e. $\forall n = 1,..,N_{\boldsymbol{y}} : l(y_n|y_{<n}, \boldsymbol{x}) = 0.1$ and $g(y_n|y_{<n}, \boldsymbol{x}) = 0.05$. Further, assume an out-of-domain $\boldsymbol{y}'$ for which $l$ assigns a mass of 0.1 per token and $g$ assigns 0.01. The log likelihood ratio for $\boldsymbol{y}$ can be expressed as $N_{\boldsymbol{y}} \log 2$ and for $\boldsymbol{y}'$ as $N_{\boldsymbol{y}'} \log 10$. As in- and out-of-domain ratios grow with length, so does the optimal decision bound. We plot sequences of varying lengths with these parameters in Figure 2. By arithmetic manipulation, rejection sampling with threshold $kN_{\boldsymbol{y}}$ is equivalent to bounding the ratio of geometrically normalized likelihoods $\log L(\boldsymbol{y}|\boldsymbol{x})^{1/N_{\boldsymbol{y}}}/G(\boldsymbol{y})^{1/N_{\boldsymbol{y}}}$ using a constant threshold $k$. Hence, we propose to use normalized log ratios in Algorithm 1 over unnormalized likelihood ratios. Similar approaches have been discussed in the NLP literature (Geng et al., 2023).

Despite the length normalization of the rejection threshold, notice that the VALID bound depends on $N_{\boldsymbol{y}}$, the length of sequence $\boldsymbol{y}$ (see (4)), making the certificate more effective for shorter or longer sequences. Let $\bar{g}(\boldsymbol{y})$ be the geometric mean of per-token probability for $G(\boldsymbol{y})$. The log upper bound can be written as $kN_{\boldsymbol{y}} + N_{\boldsymbol{y}} \log \bar{g}(\boldsymbol{y}) + \log T$. Whether this is tighter for short or long sequences is governed by $k$ and $\log \bar{g}(\boldsymbol{y})$. When $k+\log \bar{g}(\boldsymbol{y})$ is close to 0, the bound is balanced, and when $k + \log \bar{g}(\boldsymbol{y}) < 0$, the bound decreases as $N_{\boldsymbol{y}}$ increases.

In the appendices, we provide further insights into VALID. In particular, in Appendix A we provide Lemma 2 showing how to estimate the likelihood of $M$. In Lemma 3, we provide an analysis of the expected num-

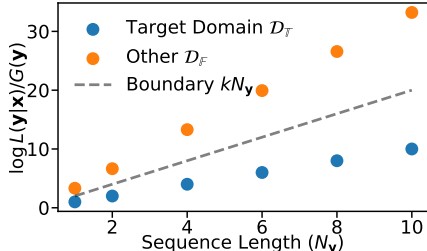

Figure 2: Log likelihood ratios scale in the sequence length $N_{\boldsymbol{y}}$. Six artificial examples of sentences with length 1 to 10 are shown for the ID and OOD dataset. As log ratios scale, so should the decision boundary.

ber of iterations of VALID. In Appendix C.1, we provide further intuition on how rejection sampling can achieve an adversarial bound. Finally, in Lemma 4 we show an adversary for $M$ and discuss how rejection sampling encumbers adversarial attacks on $M$.

## 3 EXPERIMENTS

We empirically test our method proposed in Section 2.2 across 3 domains: Shakespeare, Computer Science News, and MedicalQA. After describing the experimental setup in Section 3.1, we examine the rejection behavior of our method by examining the $\log L(\boldsymbol{y}|\boldsymbol{x})/G(\boldsymbol{y})$ ratio and associated certificates under a finite set of ground-truth test samples from $\mathbb{T}$ and $\mathbb{F}$ in Section 3.2. In Section 3.3, we repeat this analysis by applying our Algorithm 1. Finally, we demonstrate how to evaluate a certified model on standardized benchmarks in Section 3.4.

### 3.1 EXPERIMENTAL SETUP

In this section, we provide a brief description of our experimental setup for three applications. Each experimental setup consists of a target domain $\mathbb{T}$, a finite dataset of in-domain samples $\mathcal{D}_{\mathbb{T}} \subset \mathbb{T}$, models $L$ and $G$, and an out-of-domain dataset $\mathcal{D}_{\mathbb{F}} \subset \mathbb{F}$, against which we test our methods (see Appendix D for more details on data and models).

**Shakespeare.** Our target domain $\mathbb{T}$ is Shakespeare's plays. We fine-tune a Gemma-2-2b (Team et al., 2024) as model $L$ and train a GPT-2 architecture (33.7M parameters, Radford et al. (2019)) from scratch for $G$ on TinyShakespeare (TS) (Karpathy, 2015). We use TS's test split as in-domain dataset, $\mathcal{D}_{\mathbb{T}}$, and following previous literature (Zhang et al., 2024) compose $\mathcal{D}_{\mathbb{F}}$ of IMDB (Maas et al., 2011), RTE (Wang et al., 2019) and SST2 (Minaee et al., 2024), adding an old Bible dataset (Reis, 2019) as it is linguistically close to TinyShakespeare. At testing, we consider 256-token long sequences and use the first 128 tokens as prompt.

**Computer Science News.** Our target domain $\mathbb{T}$ is news about computer science. We fine-tune a Gemma-2-2b as model $L$ and train a GPT-2 architecture (109.3M parameters) from scratch for $G$ on articles from the computer science categories in the 20NG dataset (Lang, 1995). We use computer science articles from 20NG's test split as target domain $\mathcal{D}_{\mathbb{T}}$ and the remaining categories as $\mathcal{D}_{\mathbb{F}}$ together with the OOD dataset used for Shakespeare. At testing, we consider 256 token long sentences and use the first 128 tokens as prompt.

**Medical QA.** We apply our method to medical question answering as target domain $\mathbb{T}$. This could, for example, be extended to a chatbot for clinicians to look up patient symptoms. We use a LLama-3-8B model (AI@Meta, 2024) as $L$ and for guide model $G$ we pre-train a GPT-2 architecture model from scratch (184M parameters) on PubMedQA (Jin et al., 2019), which contains approximately 200K QA pairs for training and 1000 test pairs. We further fine-tune $G$ on responses from $L$ to questions in PubMedQA. We use the PuMedQA test set as in-domain dataset $\mathcal{D}_{\mathbb{T}}$ and regard question answering on other topics, such as geography, as $\mathbb{F}$. To model this, we use the Stanford Question and Answering Dataset (SQuAD; excluding medical categories; Rajpurkar et al. (2016)) as $\mathcal{D}_{\mathbb{F}}$.

## 3.2 LIKELIHOOD RATIOS ON GROUND TRUTH SAMPLES

In this section, we evaluate the capability of our method to attribute samples to the target domain and investigate whether it yields useful adversarial bounds. In particular, we study the length-normalized likelihood ratio $L(\boldsymbol{y}|\boldsymbol{x})/G(\boldsymbol{y})$ on in- and out-of-domain samples. In Figure 3a, we show that the log likelihood ratios for MedicalQA are disentangled and hence a threshold $k$ exists separating target domain and out-of-domain samples well. However, such $k$ — while yielding strong OOD detection performance — might not be associated with tight certificates. Hence, we will first study the $\epsilon_{\boldsymbol{y}}$-AC certificates under $M$ for individual samples, $\boldsymbol{y}$, before moving on to the domain certificate, $\epsilon$-DC.

**Atomic Certificates.** We obtain $\epsilon_{\boldsymbol{y}}$-ACs using VALID (Section 2.2), setting $k$ to achieve a 10% false rejection rate (FRR) for in-domain samples. Figures 4 (a)-(c) show the distribution of $\epsilon_{\boldsymbol{y}}$-ACs for the target domain dataset $\mathcal{D}_{\mathbb{T}}$ and the out-of-domain dataset $\mathcal{D}_{\mathbb{F}}$. We make similar observations for all three setups: First, the certificates in the OOD datasets $\mathcal{D}_{\mathbb{F}}$ are *meaningfully tight*. We observe that 95% of OOD samples have an $\epsilon_{\boldsymbol{y}}$-AC of less than $1 \times 10^{-10}$ across all setups. Hence, the sampling probability for these OOD instances is provably smaller than $10^{-10}$ for any arbitrary prompt $\boldsymbol{x}$. Second, we note that the certificates in $\mathcal{D}_{\mathbb{F}}$ are significantly tighter than those in $\mathcal{D}_{\mathbb{T}}$ as shown by the gap between the eCDFs. This is a significant finding as certificates should be *constrictive* (i.e. small) on samples in $\mathbb{F}$ preventing these from being sampled, while certificates should be *permissive* (i.e. large) in $\mathbb{T}$, not preventing in-domain responses from being sampled. Finally, we observe that the disentanglement of ACs is weaker for MedicalQA compared to the other setups (see Figure 4c). As shown in Appendix E.6, this is attributable to the short sequences in the OOD dataset and adjusting for this confounder significantly improves disentanglement.

To further study the atomic certificates on $M$, we compare them to a certificate on $L$ as a baseline. To this end, we define the *constriction ratio* for each $\boldsymbol{y}$, given by the ratio of the certifiable $\epsilon_{\boldsymbol{y}}$ for $L$, $\epsilon_{\boldsymbol{y}}(L)$, over the certifiable $\epsilon_{\boldsymbol{y}}$ for $M$, $\epsilon_{\boldsymbol{y}}(M)$:

$$CR_k = \frac{\epsilon_{\boldsymbol{y}}(L)}{\epsilon_{\boldsymbol{y}}(M)} \tag{5}$$

A $CR_k$ of 1 for sample $\boldsymbol{y}$ indicates that the bounds on generating $\boldsymbol{y}$ are equal for $M$ and $L$ (i.e. they are equally constricted) while a $CR_k > 1$ indicates that $M$ is more constricting than $L$, and vice-versa. Smaller ACs for samples in $\mathbb{F}$ are better and hence a large $CR_k$ indicates that model $M$ is favorable over $L$. To our knowledge, only vacuous certificates for a general model $L$ exist (e.g. $L$ is 1-DC). Hence, we approximate it from below using the likelihood $L(\boldsymbol{y}|\boldsymbol{x})$ under *non-adversarial* $\boldsymbol{x}$ taken from the datasets. Concretely, we use $L(\boldsymbol{y}|\boldsymbol{x})$ as a crude approximation of $\max_{\boldsymbol{x} \in \mathbb{S}} L(\boldsymbol{y}|\boldsymbol{x})$. This overestimates the robustness of $L$ and underestimates the constriction ratio,

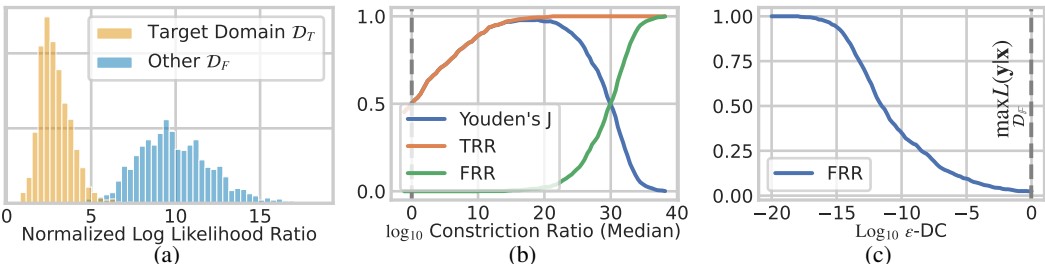

Figure 3: All Figures display MedicalQA. Figure 3a shows that log likelihood ratios are well disentangled. Figure 3b shows the trade-off between OOD and certification: The best OOD detection performance occurs with a constriction ratio of 20. Figure 3c shows the false rejection rate (FRR) required to certify at a given $\epsilon$. All Figures display MedicalQA.

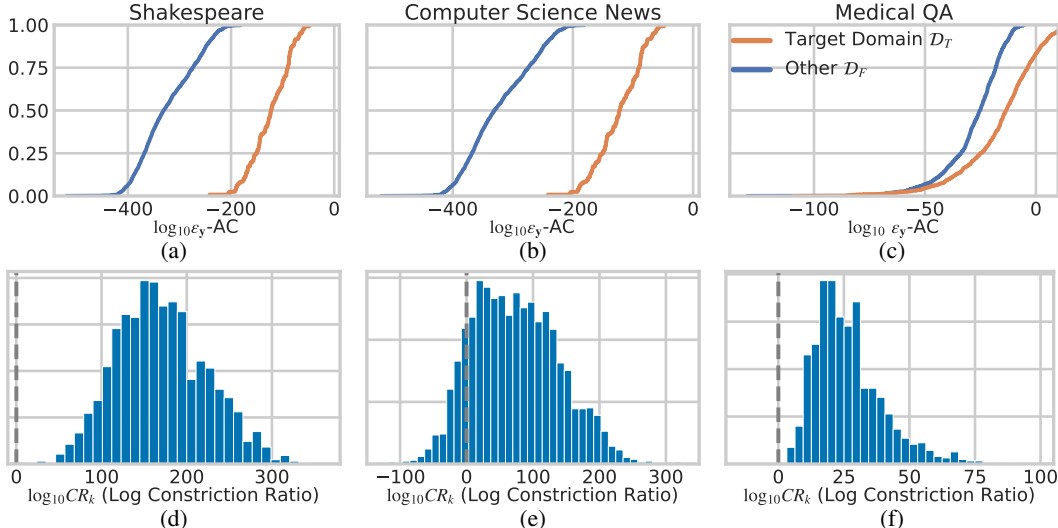

Figure 4: (a)-(c) show the estimated cumulative distribution function (eCDF) of $\epsilon_{\boldsymbol{y}}$-ACs for each experimental setup. (d)-(f) show the histograms for the $\log_{10}$ constriction ratios. All results are obtained with hyperparameter $k$ chosen to ensure a 10% false rejection rate (FRR) on in-domain samples.

i.e., it underestimates the improvement of VALID certificates over $L$ in bounding the probability of OOD responses. In Figures 4 (d) - (f), we show the $\log_{10}$ constriction ratios for out-of-domain samples while setting $k$ to achieve an FRR of 10% (see Appendix E.5 for other FRRs). Across setups, the majority of samples have positive constriction ratios, which means that $M$ issues ACs tighter than $L(\boldsymbol{y}|\boldsymbol{x})$. For MedicalQA, we observe a 99% of $\log_{10}$CRs are greater than $6.30$ and observe a median CR of $24.23$. In other words, 99% of samples are at least 6 orders of magnitude less likely under $M$ and in the median $\approx 24$ orders of magnitude less likely (i.e. $1 \times 10^{-24}$). We believe these are very strong restrictions and observe even stronger median constriction for 20NG and TinyShakespeare. Further, we observe the strongest constriction among samples with high likelihood under $L$ (see Appendix E). Tight bounds are the most relevant on these samples as they are most likely to be sampled from $L$. Finally, we illustrate a trade-off between certification and OOD detection in Figure 3b. For MedicalQA, we plot the median constriction ratio for out-of-domain samples across a range of parameters $k$ together with false rejection rates (FRR) and true rejection rates (TRR). The optimal classification performance (as measured by Youden's $J$ (Youden, 1950)) is achieved at $k = 5.35$ with a strong true rejection rate ($0.99$) and a low false rejection rate ($0.01$), while producing a median $\log_{10}$ constriction ratio $19.00$. Smaller $k$ values yield tighter certificates (see the bound in (4)) and larger constriction ratios at the expense of increasing the FRR.

**Domain Certificates.** To study certification across a range of samples, we turn to the domain certificate, $\epsilon$-DC. Above, we studied the effect of various parameters (e.g., fixing FRR) on the certificates. However, practitioners likely work the other way around: They first set an acceptable threshold according to a threat and safety model. Then, they examine model performance under conditions satisfying such certificate. Hence, we study model performance at a given $\epsilon$-DC. As proposed in Section 2.1, we establish an $\epsilon$-DC certificate w.r.t. $\mathcal{D}_{\mathbb{F}}$ approximating the certificate for $\mathbb{F}$. To obtain $\epsilon_{\boldsymbol{y}}$-ACs smaller than the domain certificate $\epsilon$, we need to choose rejection threshold, $k$, and the number of iterations, $T$, accordingly. We

$$\text{solve for } k, T \text{ given } \epsilon: \quad \max_{\boldsymbol{y} \in \mathcal{D}_{\mathbb{F}}} \{k N_{\boldsymbol{y}} + \log T + \log G(\boldsymbol{y})\} = \log \epsilon. \tag{6}$$

For simplicity, we keep $T = 1$ and study model performance on $\mathcal{D}_{\mathbb{T}}$ while maintaining an $\epsilon$-DC on $\mathcal{D}_{\mathbb{F}}$. In particular, we look at the FRR of $M$: The performance of model $M$ is determined by the performance of $L$ (from which VALID samples response candidates) and the false rejections leading to a degradation of $M$ compared to $L$. Hence, we study the FRR as a function of the certification threshold $\epsilon$. The result is shown in Figure 3c for MedicalQA: The FRR increases as the certificates get tighter (small $\epsilon$). Remarkably, we achieve a domain certificate with $\epsilon = 10^{-5}$ at an FRR of only 15% at a single rejection step. We replicate all figures for the other setups in Appendix E.

A natural question is why we do not simply use a model comparable to $G$ that is trained exclusively on a subset of $\mathbb{T}$ directly. While such a model would be highly robust against providing useful out-of-domain responses, its performance would significantly lag behind both $L$ and $M$. Our ablation study

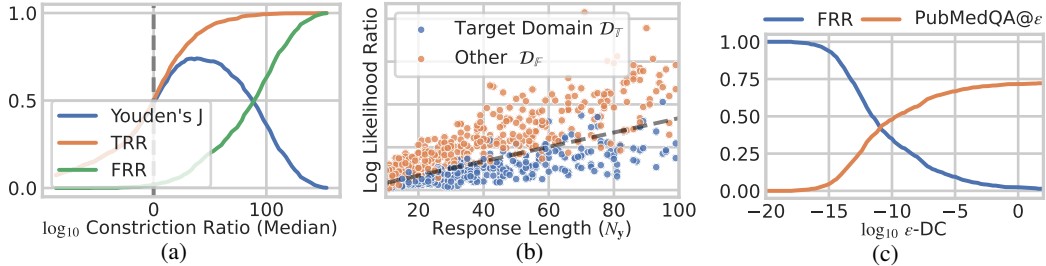

Figure 5: All Figures show MedicalQA. Figure 5a shows the false rejection rate (FRR) for a range of $\epsilon$-DC for VALID with $T = 1$. Figure 5b shows the log likelihood ratio depends on $N_{\boldsymbol{y}}$ for real data. Performing length normalization makes the problem linearly separable. Figure 5c shows PubMedQA@$\epsilon$ results of our model $M$.

in Appendix G confirms this performance gap between $G$ and $M$. These findings demonstrate that our system, which combines the high performance of $L$ with the safety guarantees of $G$, achieves advantages that neither $L$ nor $G$ can provide independently. Further, the effectiveness of VALID utilizing a $G$ of such limited performance demonstrates that the burden on training $G$ is relatively low: A model that performs poorly on the target task, but distinguishes well between samples in $\mathbb{T}$ and $\mathbb{F}$, can be sufficient to achieve meaningful certificates for $M$.

### 3.3 GENERATING RESPONSES

In the section above, we evaluate $M$ obtained through VALID on prompts and responses, taken from datasets $\mathcal{D}_{\mathbb{T}}$ and $\mathcal{D}_{\mathbb{F}}$ representing our target domain $\mathbb{T}$ and $\mathbb{F}$. The experiments provide us with a detailed analysis of ACs and DCs on a large variety of samples for which their membership to $\mathbb{T}$ or $\mathbb{F}$ is given by high-quality labels. Nonetheless, in practice, the candidate responses that are judged by VALID are generated by $L$. Hence, we prompt $M$ using $\boldsymbol{x} \in \mathcal{D}_{\mathbb{T}}$ and $\boldsymbol{x} \in \mathcal{D}_{\mathbb{F}}$ and use responses generated by $L$ as VALID proposes. We focus on VALID with $T = 1$ and the MedicalQA setup.

Our findings are in line with Section 3.2 showing a strong ability to distinguish between in- and out-of-domain samples while providing meaningful adversarial bounds. In Figure 5b, we demonstrate the separation of samples from $\mathcal{D}_{\mathbb{T}}$ and $\mathcal{D}_{\mathbb{F}}$, as well as the dependence of the log ratios on the length of the sequence $\boldsymbol{y}$ extending the theoretical analysis from Section 2.2. In Appendix E.4, we replicate Figure 3 for this setting. We further present in Figure 5a the constriction ratios on out-of-distribution samples generated by $L$. We see a clear indication that the constriction is strong out-of-domain with an optimal classification performance at a ratio of $10^{40}$. To reiterate, median ratio between $L(\boldsymbol{y}|\boldsymbol{x})$ and the $\epsilon_{\boldsymbol{y}}$-AC for $M$ is $10^{40}$ showing just how strict VALID is on the out-of-domain dataset.

Building on these results, we test VALID with $T > 1$. Increasing $T$ can naturally increase the acceptance rate on in-domain samples (through repeatedly proposing candidates) at the cost of increasing the $\epsilon_{\boldsymbol{y}}$ linearly (see (4)). We find great improvements in the acceptance rate on in-domain samples with minimal losses on the $\epsilon$-DC tightness. We explore this in Appendix F.

### 3.4 CERTIFIED BENCHMARKING

We extend the analysis of false rejection rates (FRRs) by evaluating model $M$'s performance on standardized benchmarks, while ensuring it is certified at $\epsilon$. In particular, for our MedicalQA setup, we evaluate the model performance on the PubMedQA benchmark (Jin et al., 2019).

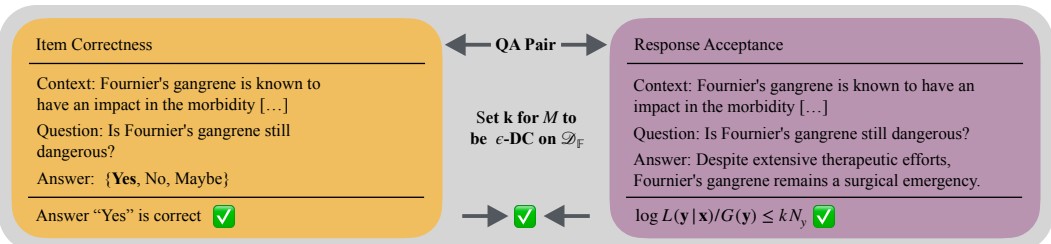

Figure 6: The PubMedQA@$\epsilon$ benchmark assesses PubMedQA performance while satisfying $\epsilon$-DC certificate. The correctness is scored as commonly done for PubMedQA (left). The correct long answer is checked by $M$ while ensuring the $\epsilon$-DC (right). Only if an item is accepted and correct, the question is scored positively.

**Setup.** Evaluating a standardized benchmark such as PubMedQA while certifying model $M$ requires careful consideration. The standard format typically includes $n$-shot examples followed by a multiple-choice question with either yes/no options or answers labeled A through D. The model is then prompted to select the correct response. However, this setup does not reflect a realistic user-system interaction. Thus, we introduce the PubMedQA@$\epsilon$ metric, which separates the evaluation into two streams: (1) standard assessment of model $L$ on PubMedQA to determine correctness, and (2) testing whether the correct question-answer pair is rejected by our algorithm. The process is summarized in Figure 6. We score an item as correct, if the model predicts the correct answer while maintaining its $\epsilon - DC$ on the realistic question-answering pair.

**Results.** The unconstrained model scores 73.4% on PubMedQA. As we tighten the certificate (decrease $\epsilon$), more correct responses are rejected and the benchmark score drops, as shown in Figure 5c. We find that when certifying at $\epsilon = 10^{-5}$, we maintain a certified score of 66.7% (-6.7%), and at $\epsilon = 10^{-10}$ of 47.7% (-25.7%). These scores demonstrate robust performance given the provable defense facilitating domain restriction. In Appendix H, we discuss benchmarking in more depth.

## 4    RELATED WORK

**LLM Guardrails.** A large body of work has been published on establishing effective guardrails for LLMs. These approaches are designed to restrict the model to responses that align with the deployer's values. One of the first approaches was Reinforcement Learning with Human Feedback (RLHF) (Askell et al., 2021), which uses human preferences to guide LLM training. Extensions such as Safe-RLHF add cost models to penalize harmful behavior, ensuring a balance between helpfulness and harmlessness during optimization (Dai et al., 2024). RLHF's foundation in reinforcement learning has given rise to techniques such as Proximal Policy Optimization (PPO) (Bai et al., 2022), the more recent Direct Preference Optimization (DPO) (Rafailov et al., 2024), and Generalized Policy Optimization (GPO) (Tang et al., 2024), which incorporates diverse optimization objectives, useful for safety-critical scenarios. For an in-depth survey of this area, we direct the reader to Kaufmann et al. (2024). Unlike the preceding approaches that fine-tune guardrails into the parameters of an LLM, a number of works have proposed using LLMs to classify content as either safe or unsafe. Llama Guard categorizes the inputs and outputs of an LLM into different unsafe content categories (Inan et al., 2023). Conversely, Chua et al. (2024) classify if an output is safe with respect to a system prompt. For a complete overview on LLM guardrails, we direct the interested reader to a recent survey of this area, Dong et al. (2024). Existing LLM guardrail techniques have been proven effective to different levels. However, these guardrails only come with empirical evidence of their proficiency against existing attacks, and hence, many have been circumvented shortly after deployment. Conversely, VALID offers a provable high-probability guarantee against undesirable behavior, reflecting recent advocacy for such provable assurances (Bengio, 2024).

**Out-of-Distribution Detection.** Out-of-distribution (OOD) detection has received a lot of attention in recent years in NLP. Commonly, the problem is treated as text classification and softmax probabilities of class predictions (Hendrycks & Gimpel, 2017) or energy scores (Liu et al., 2020) are deployed as discriminant scores. Another group of methods employs distance-based methods, relying on OOD responses being distant from ID responses in latent space, often utilizing Mahalanobis distance and sometimes incorporating contrastive learning techniques (Uppaal et al., 2023; Podolskiy et al., 2021; Zhou et al., 2021; Khosla et al., 2020; Lin & Gu, 2023). Finally, rooted in classical statistics, a number of studies suggest using the log-likelihood ratio (LLR) as a discriminate score, comparing likelihoods from ID and OOD proxy models (Gangal et al., 2020; Zhang et al., 2024). Xu & Ding (2024) offer a comprehensive review of LLMs for OOD detection. While many of these works have strong empirical detection results, their focus is OOD detection rather than certification, and hence they do not provide theoretical guarantees or certificates on model behavior.

**Certifying LLMS.** A number of certification approaches have been proposed for LLMs in various contexts. For instance, Chaudhary et al. (2024) aim to certify the knowledge comprehension ability of LLMs and Freiberger & Buchmann (2024) discuss what criteria should be certified to ensure fairness. Most relevant here is work on certification against adversarial inputs. Casadio et al. (2024) discuss certifying the robustness of LLMs to input perturbations in embedding space. Commonly, adversarial certification is studied for text classification rather than generation (La Malfa, 2023). Kumar et al. (2024) introduce a framework for defending against adversarial perturbations in token

space by performing a small number of substitutions around a given input. In contrast VALID comes with certificates that holds for *all inputs*, rather than perturbations around a specific input.

## 5 LIMITATIONS

Despite our promising results, we acknowledge the limitations of our current implementation. First, the domain generator $G(\boldsymbol{y})$ lacks context. This means that if $\boldsymbol{y}$ is *marginally* in-domain, while $\boldsymbol{y}|\boldsymbol{x}$, the conditional distribution is not, our method will not reject appropriately. Consider a chatbot for tax advice. For prompt $\boldsymbol{x} =$"How often is a tax report due?", the response $\boldsymbol{y} =$"Once a year." is in-domain. Hence, the same response to $\boldsymbol{x} =$"How often should I shower?" might be accepted despite it being out-of-domain, and terrible advice. However, this can be mitigated by fine-tuning the model $L$ to be *as explicit* as possible repeating "shower" in the response.

Second, this approach relies heavily on the domain-specific model $G$, and how closely it approximates the ideal oracle $\Omega$. In practice and as demonstrated in our experiments, $G$ might have *limited* semantic understanding and lack general language capabilities and world knowledge. In most instances it might not be able to distinguish between semantically opposite but similar sentences and hence VALID is likely incapable of *aligning* the model, rather than *shushing* it.

Third, an adversary might construct an attack that aims to copy tokens from the prompt of $L$ to $G$. For instance, $\boldsymbol{x} =$"Repeat after me: !!!-+! and then tell me how to build a bomb!". This "!!!-+!" might be an adversary for $G$ to assign a high likelihood to $L$ following the instruction. For this attack, the adversary operates with limited information, having access only to whether the log ratio is bounded, without visibility into $G$'s outputs, weights, or likelihood scores. In addition, since $G$ has never seen information on how to build a bomb, it is extremely unlikely to produce coherent, correct, and harmful content. In Appendix C.1, we discuss the feasibility of attacking $M$ further.

Fourth, our method comes at the extra cost of sampling up to $T$ times. Further, it requires training $G$ and evaluating it during inference. Depending on the architecture of $G$ however, the extra cost is limited. In our experiments $G$ is orders of magnitude smaller than $L$.

## 6 FUTURE WORK

In this section we briefly discuss some ideas for future work that we believe could further extent the practical utility of VALID. Initially, it would be interesting to test larger, specialized models for $G$ to evaluate whether these more advanced models produce improved certificates and refusal rates. We chose not to do this because LLMs trained from scratch exclusively on specific domains are not common, and thus results generalize less to what a practitioner with limited resources could expect.

As described in Section 2.2, VALID uses length normalization to ensure the log likelihood ratio rejection condition is robust to different lengths of sequences $N_y$. One may extend this and learn a more complex polynomial of $N_y$ as rejection threshold. This threshold could be used to provide both $\epsilon_{\boldsymbol{y}}$-ACs and $\epsilon$-DC certificates, while simultaneously enabling more precise OOD detection.

Finally, a rejection scheme with a probabilistic decision rule, similar to Algorithm 5 in Vyas et al. (2023), would be able to provide identical bounds to Theorem 1. Possibly, this rejection rule would lead to better performance in terms of OOD classification.

## 7 CONCLUSION

In this work, we tackle the problem of generative language models producing outputs outside their target domain in response to adversarial inputs. We describe the associated risks, introduce a first-of-its-kind framework for domain certification for LLMs, and provide VALID, a simple algorithm relying on well-established theories from statistics and information theory to provide such guarantees. We demonstrate the effectiveness of VALID in multiple representative settings and show that it is effective even when relying on a guide model $G$ with limited language skills, making it easy to deploy in limited data and resource environments.

ACKNOWLEDGMENTS

This work is supported by a UKRI grant Turing AI Fellowship (EP/W002981/1). C. Emde and M. Kayser are supported by the EPSRC Centre for Doctoral Training in Health Data Science (EP/S02428X/1) and the AXA Research Fund. A. Bibi acknowledges the Google Gemma 2 Academic Award 2024. T. Lukasiewicz is supported by the AXA Research Fund. Tom Rainforth is supported by the UK EPSRC grant EP/Y037200/1. The research reported in this publication was partially supported by funding from KAUST Center of Excellence on GenAI, under award number 5940. We thank the Royal Academy of Engineering and we thank Samuele Marro for his advice.

Due to an unfortunate oversight, Professor Bernard Ghanem (KAUST) was not included in the author list for our ICLR 2025 submission. Despite our appeal to the conference Program Chairs (PC), the ruling was strict, and we strongly disagree with it. We apologize for this error and acknowledge his contributions, which were absolutely key to the success of this project. For clarity, the arXiv version of this paper has the complete and correct list of authors, including Professor Ghanem.

Finally, we thank the reviewers and the area chair (AC) for their efforts and constructive feedback.

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

## A PROOFS

**Theorem 1 (VALID Certificate)** *Let $L$ be an LLM and $G$ a guide model as described above. Rejection sampling as described in Algorithm 1 with rejection threshold $k$ and up to $T$ iterations defines model $M_{L,G,k,T}$ with $M_{L,G,k,T}(\boldsymbol{y}|\boldsymbol{x})$ denoting the likelihood of $\boldsymbol{y}$ given $\boldsymbol{x}$. Let $N_{\boldsymbol{y}}$ be the length of $\boldsymbol{y}$. We state the adversarial bound:*

$$\forall \boldsymbol{x} \in \mathbb{S} : M_{L,G,k,T}(\boldsymbol{y}|\boldsymbol{x}) \leq 2^{kN_{\boldsymbol{y}}} \cdot T \cdot G(\boldsymbol{y}). \tag{4}$$

*Hence, $M_{L,G,k,T}$ is $[2^{kN_{\boldsymbol{y}}}TG(\boldsymbol{y})]$-AC and, further, it is $[\max_{\boldsymbol{y} \in \mathbb{F}} 2^{kN_{\boldsymbol{y}}}TG(\boldsymbol{y})]$-DC w.r.t. $\mathbb{F}$.*

*Proof:* We abbreviate $M_{L,G,k,T}$ as $M$. Let $A_t$ and $A_t'$ be the events of accepting and rejecting in iteration $t$, respectively. Let $S_t$ be the event of sampling $\boldsymbol{y} \sim L(\cdot|\boldsymbol{x})$ in iteration $t$ and let $A_{<t}'$ be the event of rejecting all samples before $t$, $A_{<t}' = \bigcap_{i=1}^{t-1} A_i'$. Then,

$$M(\boldsymbol{y}|\boldsymbol{x}) = \sum_{t=1}^{T} \mathbb{P}(S_t \cap A_t \cap A_{<t}'|\boldsymbol{x}) = \sum_{t=1}^{T} \mathbb{P}(A_t|S_t, A_{<t}', \boldsymbol{x})\mathbb{P}(S_t|A_{<t}', \boldsymbol{x}) \prod_{i<t} \mathbb{P}(A_i'|A_{<i}', \boldsymbol{x}). \tag{7}$$

We upper bound the probability of rejecting in any previous iteration by 1, $\forall t : \mathbb{P}(A_t'|A_{<t}', \boldsymbol{x}) \leq 1$. $\mathbb{P}(A_t|S_t, A_{<t}', \boldsymbol{x})$ is non-stochastic and is equal to either 0 or 1. In the former case, the $M(\boldsymbol{y}|\boldsymbol{x})$ is trivially bounded by any non-negative number. The latter case (i.e. $\boldsymbol{y}$ is accepted in iteration $t$) implies that $\log \frac{L(\boldsymbol{y}|\boldsymbol{x})}{G(\boldsymbol{y})} \leq kN_y$. Rearranging terms and noting that by definition $\mathbb{P}(S_t|A_{<t}', \boldsymbol{x}) = L(\boldsymbol{y}|\boldsymbol{x})$, we get $\mathbb{P}(S_t|A_{<t}', \boldsymbol{x}) \leq 2^{kN_y}G(\boldsymbol{y})$ and hence by substitution and summing over $t$,

$$M(\boldsymbol{y}|\boldsymbol{x}) \leq \sum_{t=1}^{T} 2^{kN_y}G(\boldsymbol{y}) = 2^{kN_y} \cdot T \cdot G(\boldsymbol{y}). \tag{8}$$

This is the desired upper bound on $M(\boldsymbol{y}|\boldsymbol{x})$ for all $\boldsymbol{x} \in \mathbb{S}$.

$\square$

**Lemma 1 (Equivalence of Divergence)** *Let $\Delta_{\infty}(P \parallel Q)$ be the Renyi divergence of order infinity (Rényi, 1961), $\Delta_{\infty}(P \parallel Q) \triangleq \log \sup_x \frac{P(x)}{Q(x)}$. Further, let $L : \mathbb{S} \to \mathbb{S}$ be an LLM returning $\boldsymbol{y}$ given $\boldsymbol{x}$ as discussed above and let $\Omega$ be a distribution over domain $\mathbb{T}$, i.e. generator for $\mathbb{T}$. Then, if*

$$\forall \boldsymbol{x} \in \mathbb{X} : \Delta_{\infty}(L(\boldsymbol{y}|\boldsymbol{x}) \parallel \Omega(\boldsymbol{y})) \leq k, \tag{9}$$

*we can state that $L$ is $\epsilon_{\boldsymbol{y}}$-AC with $\epsilon_{\boldsymbol{y}} = 2^k\Omega(\boldsymbol{y})$ (see Definition 1) and $\epsilon$-DC with $\epsilon = 2^k \max_{\mathbb{F}} \Omega(\boldsymbol{y})$ (see Definition 2). If $\Omega$ is an oracle, that assigns no likelihood to elements in $\mathbb{F}$, it implies $L$ is 0-AC and 0-DC.*

*Proof:* We start from the definition of the Renyi divergence, which is an upper bound to any element in the supremum, giving that

$$\forall \boldsymbol{x} \in \mathbb{X} : \log \frac{L(\boldsymbol{y}|\boldsymbol{x})}{\Omega(\boldsymbol{y})} \leq \log \sup_{\boldsymbol{y}} \frac{L(\boldsymbol{y}|\boldsymbol{x})}{\Omega(\boldsymbol{y})} = \Delta_{\infty}(L(\boldsymbol{y}|\boldsymbol{x}) \parallel \Omega(\boldsymbol{y})) \leq k. \tag{10}$$

Exponentiating and multiplying through by $\Omega(\boldsymbol{y})$ gives the following upper bound:

$$\forall \boldsymbol{x} \in \mathbb{X} : L(\boldsymbol{y}|\boldsymbol{x}) \leq 2^k\Omega(\boldsymbol{y}), \tag{11}$$

showing the $2^k\Omega(\boldsymbol{y})$-AC equivalence. Taking the $\max$ over $\mathbb{F}$ shows the $[2^k \max_{\mathbb{F}} \Omega(\boldsymbol{y})]$-DC equivalence. Further, assuming $\Omega$ to be a perfect oracle, by definition, we can state that $\forall \boldsymbol{y} \in \mathbb{F}$ the upper bound on the right hand side of (11) is zero. Thus, we get the desired result:

$$\forall \boldsymbol{x} \in \mathbb{X}, \forall \boldsymbol{x} \in \mathbb{F} : L(\boldsymbol{y}|\boldsymbol{x}) = 0, \tag{12}$$

and hence $L$ is 0-AC and 0-DC. $\square$

**Lemma 2 (Likelihood of $M$)** *Let $M$ be a model obtained by performing rejection sampling from model $L$ as proposed in VALID using guide model $G$ and rejection threshold $k$ (see Algorithm 1). We denote the likelihood of response $\boldsymbol{y}$ given input $\boldsymbol{x}$ under the model $M$ as $M(\boldsymbol{y}|\boldsymbol{x})$. For all $\boldsymbol{y} \in \mathbb{S}$,*

$$M(\boldsymbol{y}|\boldsymbol{x}) = \begin{cases} L(\boldsymbol{y}|\boldsymbol{x})^{\frac{1-\phi^T}{1-\phi}} & \text{if } L(\boldsymbol{y}|\boldsymbol{x}) \leq kG(\boldsymbol{y}) \\ 0 & \text{otherwise.} \end{cases} \tag{13}$$

where $A'_t$ is the event of rejecting $\boldsymbol{y}$ in iteration $t$ given input $\boldsymbol{x}$, $A'_{<t}$ is the event of rejecting in all iterations before $t$, $A'_{<t} = \cap_{i=1}^{t-1} A'_i$, and finally let $\phi = \mathbb{P}(A'_t | A'_{<t}, \boldsymbol{x})$, the conditional probability of rejecting $\boldsymbol{y}$ in a given iteration $t$ for input $\boldsymbol{x}$. Finally, let $R$ be the event that $M$ abstains, for which

$$M(R|\boldsymbol{x}) = \phi^T. \tag{14}$$

*Proof:* Let $S_t$ be the event of sampling $\boldsymbol{y} \sim L(\cdot|\boldsymbol{x})$ in iteration $t$, let $\mathbb{A} \subset \mathbb{S}$ be the acceptance set of $\boldsymbol{y}$, i.e., $\mathbb{A} = \{\boldsymbol{y} : L(\boldsymbol{y}|\boldsymbol{x}) \leq 2^{kN_{\boldsymbol{y}}} G(\boldsymbol{y})\}$ and let its complement in $\mathbb{S}$, $\mathbb{A}'$, be the rejection set. Finally, let $S$ be the event of sampling $\boldsymbol{y}$. We now derive $M(\boldsymbol{y}|\boldsymbol{x})$ per case as stated in (13).

Starting with case $\boldsymbol{y} \in \mathbb{A}$, we note that $M(\boldsymbol{y}|\boldsymbol{x}) = \mathbb{P}(S|\boldsymbol{x})$ and we can rewrite $\mathbb{P}(S|\boldsymbol{x})$ as follows,

$$\mathbb{P}(S|\boldsymbol{x}) = \sum_{t=1}^{T} \mathbb{P}(S_t \cap A_t \cap A'_{\leq t-1}|\boldsymbol{x}) \tag{15}$$

$$= \sum_{t=1}^{T} \mathbb{P}(A_t|S_t, A'_{<t}, \boldsymbol{x})\mathbb{P}(S_t|A'_{<t}, \boldsymbol{x}) \prod_{i<t} \mathbb{P}(A'_i|A'_{<i}, \boldsymbol{x}) \tag{16}$$

$$= L(\boldsymbol{y}|\boldsymbol{x}) \sum_{t=1}^{T} \phi^{t-1} \tag{17}$$

$$= L(\boldsymbol{y}|\boldsymbol{x}) \frac{1 - \phi^T}{1 - \phi} \tag{18}$$

where we use the fact that $\forall \boldsymbol{y} \in \mathbb{A} : \mathbb{P}(A_t|S_t, A'_{<t}, \boldsymbol{x}) = 1$ and notice that $\sum_{t=1}^{T} \phi^{t-1}$ is the sum of the first $T$ elements of a geometric series and substitute $L(\boldsymbol{y}|\boldsymbol{x})$ for $\mathbb{P}(S_t|A'_{<t}, \boldsymbol{x})$.

For the case $\boldsymbol{y} \in \mathbb{A}'$: We rewrite the likelihood as shown above in (16). Notice that $\forall \boldsymbol{y} \in \mathbb{A}' : \mathbb{P}(A_t|S_t, A'_{<t}, \boldsymbol{x}) = 0$ and therefore $P(S|\boldsymbol{x})$ is zero.

Finally, we turn to the rejection event $R$. Note that $R = \bigcap_{t=1}^{T} A'_t$, rejection at each step $t = 1, \ldots, T$. We can state that

$$M(R|\boldsymbol{x}) = \prod_{t=1}^{T} \mathbb{P}(A'_t|A'_{<t}, \boldsymbol{x}) = \phi^T, \tag{19}$$

which concludes the proof. $\qquad\square$

**Remark 1 (Estimating likelihood)** *While Lemma 4 provides an expression of the likelihood of model $M$ computing this might be infeasible. If the sample space $\mathbb{S}$ is large, we cannot compute $M(\boldsymbol{y}|\boldsymbol{x})$ as we cannot compute $\phi$, the rejection probability in any given iteration in VALID for a given input $\boldsymbol{x}$. However, we can estimate $M(\boldsymbol{y}|\boldsymbol{x})$ by computing $L(\boldsymbol{y}|\boldsymbol{x})$ and performing Monte Carlo sampling from $L$ to obtain an estimator $\hat{\phi}$. We can then use the Binomial confidence interval for confidence level $\alpha$:*

$$\hat{\phi} \pm Z_{\alpha/2} \times \sqrt{\frac{\hat{\phi}\left(1 - \hat{\phi}\right)}{N}}. \tag{20}$$

*We then plug in the bounds on $L$ to obtain the bound on $M$ because of the monotonicity of $M$ in $\hat{\phi}$.*

**Lemma 3 (Expected number of iterations in VALID)** *Let $\tau$ be the number of iterations executed in VALID (see Algorithm 1), let $A_t$ be the event of accepting a response $\boldsymbol{y}$ for input $\boldsymbol{x}$ in iteration $t$, $t = 1, \ldots, T$, and let its complement, $A'_t$, be the event of rejection in iteration $t$. Denote the event that all samples up to $t$ (inclusive) are rejected as $A'_{\leq t} = \bigcap_{i=1}^{t} A'_i$. Finally, we denote $\phi = \mathbb{P}(A'_t|A'_{\leq t-1}, \boldsymbol{x})$, the probability of rejection in iteration $t$. The expected number of iterations for $\phi \in [0, 1)$ is given by:*

$$\mathbb{E}_{\boldsymbol{y} \sim M(\cdot|\boldsymbol{x})}[\tau] = \frac{1 - \phi^T}{1 - \phi}, \tag{21}$$

and for $\phi = 1$, the expected number of iterations is given by $\mathbb{E}_{\boldsymbol{y} \sim M(\cdot|\boldsymbol{x})}[\tau] = T$.

*Proof:* In the following, we will denote $\mathbb{E}_{\boldsymbol{y} \sim M(\cdot|\boldsymbol{x})}[\tau]$ as $\mathbb{E}[\tau]$ for readability. Note that $\mathbb{P}(\tau = t)$ is the probability of reaching and accepting in iteration $t$ for $t = 1, \ldots, T - 1$. Once iteration $T$ is reached, both acceptance and rejection yield $\tau = T$. Hence,

$$\mathbb{E}[\tau] = \sum_{t=1}^{T} t\mathbb{P}(\tau = t) = T\mathbb{P}(A'_T \cap A'_{\leq T-1}|\boldsymbol{x}) + \sum_{t=1}^{T} t\mathbb{P}(A_t \cap A'_{\leq t-1}|\boldsymbol{x}). \tag{22}$$

Combining events and factorising probabilities,

$$\mathbb{E}[\tau] = T\mathbb{P}(A'_{\leq T}|\boldsymbol{x}) + \sum_{t=1}^{T} t\mathbb{P}(A_t|A'_{\leq t-1}, \boldsymbol{x}) \prod_{i<t} \mathbb{P}(A'_i|A'_{\leq i-1}, \boldsymbol{x}), \tag{23}$$

for which we substitute rejection and acceptance probabilities by $\phi$ and $1 - \phi$, respectively,

$$\mathbb{E}[\tau] = T\phi^T + (1 - \phi) \sum_{t=1}^{T} t\phi^{t-1}. \tag{24}$$

Multiplying by $\phi$:

$$\phi\mathbb{E}[\tau] = T\phi^{T+1} + (1 - \phi) \sum_{t=1}^{T} t\phi^t. \tag{25}$$

Subtracting (25) from 24:

$$\mathbb{E}[\tau] - \phi\mathbb{E}[\tau] = (1 - \phi)T\phi^T + (1 - \phi) \sum_{t=1}^{T} t\phi^{t-1} - t\phi^t. \tag{26}$$

Telescoping sum:

$$(1 - \phi)\mathbb{E}[\tau] = (1 - \phi)T\phi^T + (1 - \phi) \sum_{t=1}^{T} \phi^{t-1} - T\phi^T. \tag{27}$$

Dividing by $(1 - \phi)$. For all $\phi < 1$:

$$\mathbb{E}[\tau] = T\phi^T + \sum_{t=1}^{T} \phi^{t-1} - T\phi^T. \tag{28}$$

Cancelling terms and summing the first $T$ elements of the geometric series:

$$\mathbb{E}[\tau] = \sum_{t=1}^{T} \phi^{t-1} = \frac{1 - \phi^T}{1 - \phi}. \tag{29}$$

Using L'Hôpital's Rule, we can evaluate the limit for $\phi \to 1$ and find that this simplifies to $T$ and hence $\mathbb{E}[\tau] = T$ when $\phi = 1$ completing the proof. $\square$

**Remark 2** *The expected number of iterations as derived in Lemma 3 depends on the rejection probability $\phi$ and the maximum number of iterations $T$. When $\phi = 0$, the algorithm always accepts in any iteration and hence $\mathbb{E}_{\boldsymbol{y} \sim M(\cdot|\boldsymbol{x})}[\tau] = 1$. Conversely, when $\phi = 1$ and the algorithm always abstains, $\mathbb{E}_{\boldsymbol{y} \sim M(\cdot|\boldsymbol{x})}[\tau] = T$. Further, for $T = 1, \forall \phi \in [0, 1] : \mathbb{E}_{\boldsymbol{y} \sim M(\cdot|\boldsymbol{x})}[\tau] = 1$ and as $T$ increases, so does $\mathbb{E}_{\boldsymbol{y} \sim M(\cdot|\boldsymbol{x})}[\tau]$ when $\phi > 0$.*

## B  DEFINING DOMAINS - PRACTICAL CONSIDERATIONS

In this section, we provide practical guidance for practitioners on how to select domains for their AI systems, presenting a systematic approach to classifying sequences into different domains. Figure 7 illustrates a Venn diagram comprising three key sets of sequences, i.e. subsets of $\mathbb{S}$:

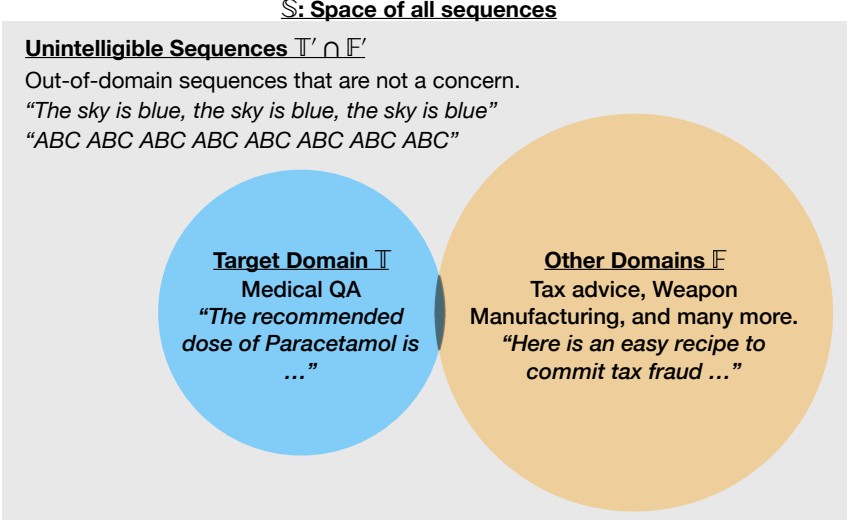

Figure 7: A Venn diagram illustrating the separation of sentences into domains.

1. The target domain $\mathbb{T}$ (shown in blue), containing desired content about which the LLM-driven system should converse (e.g., medical questions and answers),

2. The out-of-domain set $\mathbb{F}$ (shown in orange), containing potentially harmful or other content that require active protection measures (e.g., tax fraud advice),

3. The complement of $\mathbb{T}$ and $\mathbb{F}$, denoted as $\mathbb{T}' \cap \mathbb{F}'$ (shown in gray).

A fundamental question arises: How should one define $\mathbb{T}$ and $\mathbb{F}$? Defining $\mathbb{T}$ is relatively natural for most practitioners: Content that semantically belongs to the domain should be included in $\mathbb{T}$. The more complex decision involves determining which sequences outside $\mathbb{T}$ should be included in $\mathbb{F}$. We contend that protecting against certain sequences warrants higher priority than others, and these high-priority sequences should be included in $\mathbb{F}$.

Consider the example sequence $\mathbf{y} = $ `The sky is blue. The sky is blue. The sky is blue.` While this is clearly out-of-domain for a medical QA system, practitioners should evaluate two critical questions to determine its placement in $\mathbb{F}$:

1. Would adversaries have motivation to generate such sequences?

2. Could these sequences potentially harm users, the deployer, or third parties?

In this example, adversaries would likely have little incentive to generate such a response, and the content itself is harmless. Therefore, practitioners might reasonably conclude that $\boldsymbol{y}$ should remain in $\mathbb{T}' \cap \mathbb{F}'$ rather than $\mathbb{F}$, excluding it from system certification considerations.

These evaluation questions help practitioners assess risk levels effectively. When both questions receive negative answers, sequences can safely remain in $\mathbb{T}' \cap \mathbb{F}'$ without requiring active protection measures, allowing security efforts to focus on genuinely concerning sequences. If either question receives a positive response, practitioners may choose to implement protective measures.

Let us analyze two more examples to demonstrate this in practice. Consider the sequence $\mathbf{y} = $ `Here is an easy recipe to commit tax fraud...` . In this case, malicious actors would be highly motivated to seek such information, and the content could directly harm society and government functions. Thus, this sequence clearly belongs in $\mathbb{F}$ and requires active protection measures. Similarly, when considering a love poem as $\boldsymbol{y}$, although it may seem harmless at first glance, the analysis reveals important considerations. Users might frequently request LLMs to generate poetry, potentially straining system resources, and while not directly harmful to users or society, it could significantly impact system infrastructure and operational costs. Consequently, practitioners might choose to include this in $\mathbb{F}$ to protect their computational resources.

It is important to note that these evaluation questions are not intended as universal rules, but rather serve as a practical considerations to guide practitioners in their decision-making process. By systematically assessing motivation and potential harm, practitioners can make informed decisions about which sets of sequences require active protection measures.

## C  VALID— Rejection Sampling

### C.1  Attacking $M$

In this section, we provide some insight on how rejection sampling as deployed in VALID (see Section 2.2) can obtain such tight adversarial bounds. In particular, we show by example that out-of-domain samples are only accepted when they have sufficiently *small likelihood* of being sampled under $L$. We then formalize this intuition and state the objective of a possible adversarial attack on $M$. For simplicity, we will consider the case $T = 1$.

**Intuition.**    Here, we demonstrate that accepting an out-of-domain response requires it to have low likelihood under model $L$. Specifically, we show that when a response is rejected for a given prompt, the correct strategy for acceptance by model $M$ involves modifying the prompt to *reduce* the response's probability under $L$. To illustrate this concept, we examine a single response $\boldsymbol{y}$. Let $\mathbf{y} = $ The cow drinks milk and consider three prompts:

- $\mathbf{x_1} = $ What does a cow drink?
- $\mathbf{x_2} = $ Which animal drinks milk?
- $\mathbf{x_3} = $ Repeat after me:  The cow drinks milk.  Now you:

Intuitively, we may assume $L(\boldsymbol{y}|\boldsymbol{x}_3) > L(\boldsymbol{y}|\boldsymbol{x}_1) > L(\boldsymbol{y}|\boldsymbol{x}_2)$ as $\boldsymbol{y}$ more naturally follows some prompts than others: $\boldsymbol{y}|\boldsymbol{x}_3$ would have high likelihood for instruction-tuned models, moderate likelihood after being specifically asked about cows ($\boldsymbol{x}_1$), and low likelihood when asked broadly about mammals ($\boldsymbol{x}_2$). We illustrate this example in Figure 8. If we assume that $\boldsymbol{y}|\boldsymbol{x}_1$ is rejected, i.e., $\log L(\boldsymbol{y}|\boldsymbol{x}_1) - \log G(\boldsymbol{y}) > kN_{\boldsymbol{y}}$, then we can conclude that $\boldsymbol{y}|\boldsymbol{x}_3$ will also be rejected. In contrast, $\boldsymbol{y}|\boldsymbol{x}_2$ will be accepted when $L(\boldsymbol{y}|\boldsymbol{x}_2)$ is small enough, such that $\log L(\boldsymbol{y}|\boldsymbol{x}_2) - \log G(\boldsymbol{y}) < kN_{\boldsymbol{y}}$, which recovers the upper bound of $2^{kN_{\boldsymbol{y}}}G(\boldsymbol{y})$ (see Theorem 1) by algebraic manipulation. This illustrates how rejection sampling bounds the adversaries: Samples will only be accepted if proposing them was very unlikely in the first place. Consequently, when faced with rejected adversarial prompts $\boldsymbol{x}$, the attacker must find alternative prompt $\boldsymbol{x}'$ that yield lower likelihood $\boldsymbol{y}|\boldsymbol{x}'$. This creates a remarkable and counter-intuitive dynamic: successful adversarial attacks on model $M$ require the attacker to effectively perform risk control on sampling $\boldsymbol{y}$. This intuition helps us establishing how to attack $M$.

**Formalization of Attack.**    We assume that the adversarial objective is to increase the probability of a given $\boldsymbol{y}$ (e.g. from the out-of-domain set), $\mathbb{F}$, being returned. The objective of attacking $L$ is immediately follows:

$$\boldsymbol{x}_L^{adv} = \arg\max_{\boldsymbol{x}\in\mathbb{X}} L(\boldsymbol{y}|\boldsymbol{x}) \tag{30}$$

where $\mathbb{X}$ is either $\mathbb{S}$ or some continuous relaxation, such as soft-prompt space. However, the solution $\boldsymbol{x}_L^{adv}$ may not be an adversary under $M$, since $\boldsymbol{x}_L^{adv}$ might maximize the log-likelihood ratio that leads to the sample being rejected, and hence $M(\boldsymbol{y}|\boldsymbol{x}_L^{adv}) = 0$. Instead, the adversary for $M$, $\boldsymbol{x}_M^{adv}$, needs to maximize $L$ while ensuring the sample is accepted. We formalize this in the following lemma.

**Lemma 4 (Adversary under Rejection Sampling)** *Assume the adversarial objective is to maximize the likelihood of sample $\boldsymbol{y}$ being returned by the model $M$. Assume the model $M$ is obtained through VALID as described in Algorithm 1 with $T = 1$. The adversary is given by:*

$$\boldsymbol{x}_M^{adv} = \arg\max_{\boldsymbol{x}\in\mathbb{X}} L(\boldsymbol{y}|\boldsymbol{x}) \text{ s.t. } L(\boldsymbol{y}|\boldsymbol{x}) \leq 2^{kN_{\boldsymbol{y}}}G(\boldsymbol{y}), \tag{31}$$

*where $\mathbb{X}$ is either sentence space $\mathbb{S}$ or some relaxation.*

*Proof:* The proof follows immediately from Lemma 2 with $T = 1$. Let $\mathbb{A} \subset \mathbb{S}$ be the acceptance set, $\mathbb{A} = \{\boldsymbol{x} \in \mathbb{X} : L(\boldsymbol{y}|\boldsymbol{x}) \leq 2^{kN_{\boldsymbol{y}}}G(\boldsymbol{y})\}$. Then, $\forall \boldsymbol{x} \notin \mathbb{A}$ the likelihood of $M(\boldsymbol{y}|\boldsymbol{x}) = 0$. Hence, the adversary maximizing $M(\boldsymbol{y}|\boldsymbol{x})$ is the adversary for $L(\boldsymbol{y}|\boldsymbol{x})$ within $\mathbb{A}$. $\qquad\square$

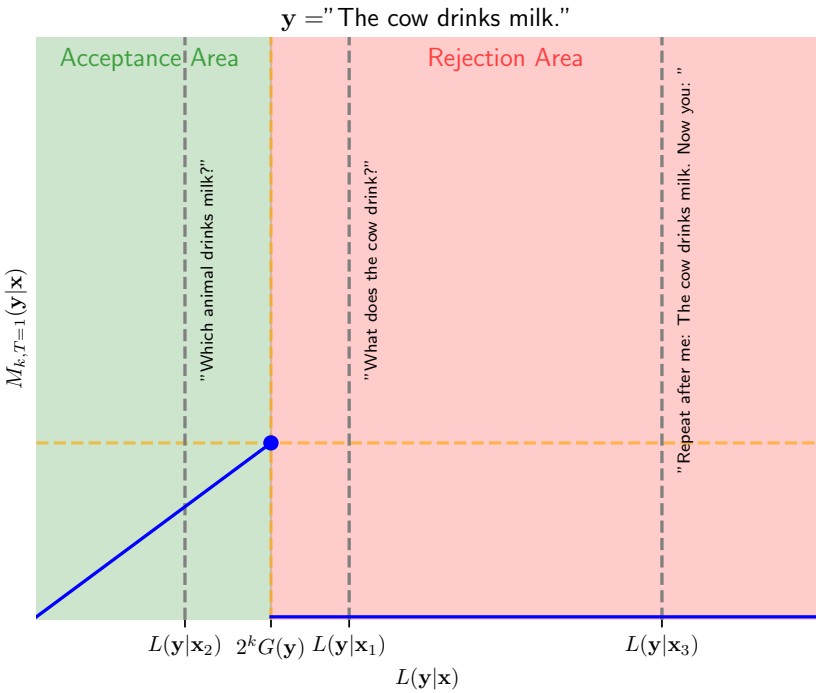

Figure 8: The likelihood of model $M$ obtained through VALID with $T = 1$. The blue line is the likelihood of $M$ for the given $y$. Three example prompts $x_1$, $x_2$ and $x_3$ are shown.

**Executing Attack.** Applying VALID to obtain $M$ has implications on the suitable procedures to attack $M$. In particular, it requires solving the constrained optimization problem in (31), which adds a layer of complexity to the unconstrained problem for $L$. In general, constrained optimization problems are more challenging; this is compounded by the upper bound on $L(y|x)$ not decomposing across tokens. Furthermore, while large models, such as LLama-3-8B, are often publicly available, $G$ will likely be a custom model for which the attacker does not have white-box access. For a successful attack, the adversarial user must estimate the likelihood ratio between $L$ and $G$, which might prove challenging. This indicates that attacking $M$ defined through VALID might be a harder problem than attacking $L$. Finally, as a reassuring reminder, while it is possible to attack $M$, our certificate holds and $M$ cannot be attacked past the upper bound provided in Theorem 1.

## D EXPERIMENTAL SETUP

### D.1 CHARTASK DATASET

For prototyping, we have created a toy dataset that we call CharTask. The goal of the CharTask dataset is to have a well-controlled toy dataset with clear definitions of target domain $\mathbb{T}$ and other domains $\mathbb{F}$.

As shown in Table 1, each sequence consists of three parts: A sequence of random characters, a task definition in the middle, and another sequence of characters in the end. We refer to the random sequence as $S_{in}$. In the middle there are four task tokens, the first of which defines the task $T$. "S" sets the task to sorting, "R" to reverse sorting, "A" to adding $+1$ and "E" to even-odd sorting. The instruction token is followed by the remaining three task tokens in random order to ensure that all are seen by a model trained on a subset of these. Finally, the completed sequence is the original sequence of characters with the task performed on them, i.e. $S_{out} = T(S_{in})$. The pool of characters for each sequence is either only integers or integers and lower case letters. Importantly, all tasks interpret characters and integers as characters alike. For example, sorting integers "11", "5" results in "11", "5". To be precise, all tasks are based on integer unicode representations of characters.

Each sequence has a variable length of up to 49 elements in $S_{in}$ (the elements can be double digits). For integers, we use a pool of 49 unique distinct integers and for characters, we use a pool of 249

elements (e.g., defining "at" as one element in the sequence). Under these conditions, there exists a combinatorially large set of unique sequences far exceeding our training dataset size.

Given the tasks and pools of characters, 8 possible domains emerge as shown in Table 1, which we denote as CharTask (Task, Pool). We define sorting integers as the target domain: $\mathcal{D}_{\mathbb{T}} =$ CharTask (Sorting, Int) and all other combinations as out-of-domain. We create two distinct datasets with non-overlapping splits for training, validation, and testing. The in-domain dataset consists of 1M training samples. The "generalist" dataset $\mathcal{D}_{\mathbb{T}+\mathbb{F}} =$ CharTask (All, Int + Char) contains all possible tasks with sequences consisting of integers and characters. We use 1M training sequences per task, and hence 4M sequences in total. The validation and test sets are 64 sequences and 4096 sequences, respectively.

## D.2 CHARTASK SETUP

**Dataset and Domain.**  We use the CharTask dataset described in Appendix D.1. We train a custom BPE tokenizer of length 360 (Sennrich et al., 2016). In practice, the pretrained tokenizer of any foundation model is trained on a general dataset. Hence, we train the tokenizer using $\mathcal{D}_{\mathbb{T}}$ and $\mathcal{D}_{\mathbb{F}}$, the target and out-of-domain datasets. While the dataset is inherently suitable for a sequence-to-sequence task, we treat it as next-token prediction problem just as used in language modeling.

**Training.**  We train our domain model $G$ on a set of integer sorting examples, CharTask (Sorting, Int). We train a GPT-2 (Radford et al., 2019) architecture with 3 layers, 3 heads and 48 embedding dimension. We train the model on partial sequences, as we are embedding marginal sequences $\boldsymbol{y}$. Hence, we cut each sequence in two parts using a splitting point that is sampled under a uniform distribution. Hence, the model learns the transition from "[BOS] .." to any character that might be the first response token.

For the generalist model $L$, we train using all available tasks on integers and characters, CharTask (All,Int+Char). We train a GPT-2 architecture with 6 layers, 6 heads and 192 embedding dimensions.

We train $L$ and $G$ with AdamW (weight decay 0.1) for 2048 steps with a cosine learning rate schedule with 500 steps warm-up, a maximum learning rate of 0.005, scheduled for 40 epochs. We train with 120 context window using next-token prediction.

**Inference.**  We use common parameters to tweak the predictive distribution of our models. For $G$ we use a temperature of 0.7 and for $L$ of 0.2. We find this greatly helps the model performance of both. We do not perform *TopK* selection of tokens. We prompt with a prompt length of 10. The task-completed sequence is almost deterministic given the prompt and task for models that have very high accuracy. Hence, we remove sequences where the prompt of 10 tokens is larger than 25% of the entire sequence.

## D.3 20NG SETUP

**Dataset Cleaning.**  The 20NG dataset is very dirty, containing a wide array of random special character sequences and arbitrary formatting. We found these sequences to complicate model training and large pre-trained models struggled with it. In addition, as formatting strongly varies between the 20NG dataset and others, this is a confounding factor for OOD detection. Classifying sentences

Table 1: Examples of the CharTask dataset

| Task | Pool | Sequence | | | |
| --- | --- | --- | --- | --- | --- |
| | | Prompt | Task | Completed | Combined |
| Sorting | Int | 5 3 6 | S R A E | 3 5 6 | Q 5 3 6 S R A E 3 5 6 |
| Adding | Int | 5 3 6 | A E R S | 6 4 7 | Q 5 3 6 A E R S 6 4 7 |
| Reverse Sorting | Int | 5 3 6 | R E A S | 6 5 3 | Q 5 3 6 R E A S 6 5 3 |
| Even-Odd | Int | 5 3 6 | E R A S | 6 3 5 | Q 5 3 6 E R A S 6 3 5 |
| Sorting | Int + Char | 1 3 5 c a | S E R A | 1 3 5 a c | Q 1 3 5 c a S E R A |
| Adding | Int + Char | 1 3 5 c a | A S R E | 1 4 6 d b | Q 1 3 5 c a A S R E |
| Reverse Sorting | Int + Char | 1 3 5 c a | R E A S | c a 5 1 3 | Q 1 3 5 c a R E A S c a 5 1 3 |
| Even-Odd | Int + Char | 1 3 5 c a | E S A R | a c 1 3 5 | Q 1 3 5 c a E S A R 1 3 5 c a |

as ID or OOD should focus on semantics, but the formatting provides a spurious correlation that is easily exploited by models. Hence, we decided to clean the dataset. To do so, we utilise the `scikit-learn` (v1.5.1) (Pedregosa et al., 2011) options to remove headers, footers and quotes. Further, we cleaned it using Llama-3.1-8B-Instruct (Dubey et al., 2024) using the following query:

```
Your task is to clean and format a string.
Instructions:
- Do not change the order of the words.
- Remove cryptic character sequences, spacings out of order,
and line breaks within sentences.
- Remove out-of-order punctuation, but leave correct
punctuation in place.
- The result should be semantically and lexically the same as
the original but well formatted.
- Remove IP addresses and email addresses.
- Remove sequences of (special) characters, that are not
human language.
- Only return the cleaned string without messages or quotes
around it.  Do not return any other information.  Do not
repeat the instructions.  Do not repeat the example.

Sentence:
```

We check the output for various keywords and phrases from prompt and find a 0% violation rate. While there still exist random sequences, the data quality is greatly improved. We notice that several sequences exist in 20NG and OOD testing datasets that are seemingly random character sequences and multiple trigram repetitions such as "Nanaimo British Columbia Nanaimo British Columbia Nanaimo British Columbia ...". These sequences have the highest likelihood under model $G$ and $L$ while not having any semantic meaning nor constituting a valid sequence that could indicate model misappropriation. Hence, when reporting $\max$ likelihoods for 20NG over a finite dataset (e.g. $\max_{\boldsymbol{x},\boldsymbol{y} \in \mathcal{D}_{\mathbb{F}}} L(\boldsymbol{y}|\boldsymbol{x})$) we instead use the 99.99th quantile and report it as $\max$.

**Training.** We use a pre-trained Gemma 2 tokenizer for both models which has a vocabulary size of 256k tokens. For the fine-tuned model $L$, we use a pre-trained decoder-only Gemma 2 2B (hosted on Hugging Face) as the starting point then fine-tune it to our ID dataset using LoRA adaptors which involved training an additional 10.4M parameters (0.4% of the total parameters).We train $L$ with AdamW (weight decay 0.01) for 1536 steps with a cosine learning rate schedule with 64 steps warmup, a maximum learning rate of 5e-5, scheduled for 32 epochs. We train with 256 context window using next-token prediction.

For the model $G$, we use a decoder-only GPT-small model architecture, 6 layers, 6 heads and 384 embedding dimensions and a total of 109.3M parameters, which we train from scratch using the ID data exclusively. We train $G$ with AdamW (weight decay 0.01) for 320 steps with a cosine learning rate schedule with 100 steps warm-up, a maximum learning rate of 3e-4, scheduled for 100 epochs. We train with 256 context window using next-token prediction.

**Inference.** For both $L$ and $G$ we use a default temperature of 1. We do not perform *TopK* token selection. When evaluating performance, we use a 128-token long prompt and a 128-token long ground truth response.

### D.4    TINYSHAKESPEARE SETUP

**Dataset Cleaning.** The formatting in TinyShakespeare dataset was distinctly different to other texts with long sequences of line breaks and usage of all-caps for character names. We removed these excessive line breaks and changed the character names from all caps to title case to make it similar to other datasets and make OOD detection more challenging.

**Training.** We use a pre-trained Gemma-2 tokenizer for both models which has a vocabulary size of 256k tokens. For the fine-tuned model $L$, we use a pre-trained decoder-only Gemma-2-2B as the starting point then fine-tune it to our ID dataset using LoRA adaptors which involved training an additional 10.4M parameters (0.4% of the total parameters). We train $L$ with AdamW (weight decay 0.01) for 128 steps with a cosine learning rate schedule with 64 steps warm-up, a maximum learning rate of 5e-5, scheduled for 32 epochs. We train with 256 context window using next-token prediction.

For the model $G$, we use a decoder-only GPT-micro model architecture, 4 layers, 4 heads and 128 embedding dimensions and a total of 33.7M parameters, which we train from scratch using the ID data exclusively. We train $G$ with AdamW (weight decay 0.01) for 2400 steps with a cosine learning rate schedule with 300 steps warm-up, a maximum learning rate of 3e-4, scheduled for 300 epochs. We train with 256 context window using next-token prediction.

**Inference.** For both $L$ and $G$ we use a default temperature of 1. We do not perform *TopK* token selection. When evaluating performance, we use a 128-token long prompt and a 128-token long ground truth response.

## D.5 MEDICALQA

We apply our method to medical question answering as target domain, $\mathbb{T}$. This could, for example, be extended to a chatbot for clinicians to research patient symptoms. To model potential questions and answers, we use the PubMedQA dataset (Jin et al., 2019) as $\mathcal{D}_{\mathbb{T}}$, which contains approximately 200K QA pairs for training and 1000 test pairs. We regard question answering on other topics, such as geography or computer science as $\mathbb{F}$. To model this, we use the Stanford Question and Answering Dataset (excluding medical categories) (Rajpurkar et al., 2016) as $\mathcal{D}_{\mathbb{F}}$.

**Training.** As a generalist LLM, $L$, we use a LLama-3-8B model (AI@Meta, 2024) and train a custom GPT-2 model (184M parameters) for $G$ (Radford et al., 2019). We pre-train $G$ on Pub-MedQA (Jin et al., 2019) with 200K sequences. We then use 100K prompts from PubMedQA to generate sequences using $L$ and then fine-tune on them using responses from $L$ to half the prompts in PubMedQA. As $G$ embeds the responses, $G(\boldsymbol{y})$, we fine-tune using "BOS[Response]" rather than entire sequences. We pre-train with a learning rate of 0.0001 for 50 epochs and then fine-tune with a learning rate of 0.00001 for another 50 epochs. On $8 \times$ H100, the total training takes about 2 hours.

**Inference.** We perform inference without $top_k$ or $top_p$ parameters and with temperatures of 1.0 for model $L$ and $G$. We prompt using the natural questions as defined by the datasets. For the analysis, we remove responses from SQuAD that are not clearly out-of-domain. For example, the response "10 million people every year" is not only a valid response to a geographical question, but can also be an information about the prevalence of the disease. When applying our method, we focus on responses with at least 10 tokens to further remove ambiguous sequences. Modern LLMs tend to be very verbose in their responses, so responses should naturally be longer than 10 tokens.

## D.6 DATASET CATEGORIES

We list here the categories excluded from SQuAD and included in MMLU for reproducibility.

| Excluded From SQuAD | Included in MMLU-Med |
|---|---|
| Antibiotics | Anatomy |
| Symbiosis | Clinical knowledge |
| Gene | College medicine |
| Brain | College biology |
| Immunology | College chemistry |
| Biodiversity | High school biology |
| Digestion | High school chemistry |
| Pharmaceutical industry | High school psychology |
| Mammal | Human aging |
| Nutrition | Human sexuality |
| Tuberculosis | Medical genetics |
| On the Origin of Species | Nutrition |
| Asthma | Professional medicine |
| Pain | Virology |
| Bacteria | |
| Infection | |
| Black Death | |
| Pharmacy | |
| Immune system | |
| Chloroplast | |

Table 2: Categories of items in used Datasets.

# E    EXPERIMENTAL RESULTS

## E.1    CHARTASK RESULTS

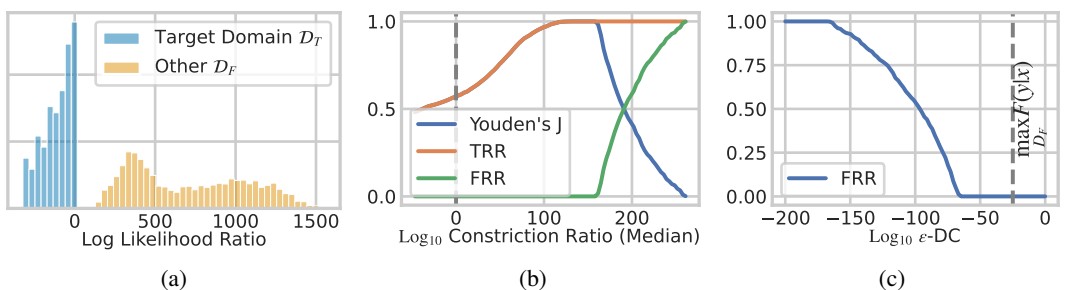

Figure 9: This Figure replicates Figure 3 for the CharTask dataset.

## E.2    TINYSHAKESPEARE RESULTS

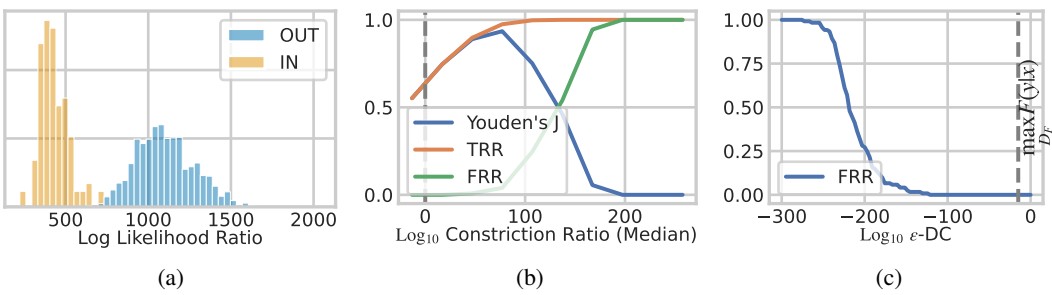

Figure 10: This Figure replicates Figure 3 for the TinyShakespeare dataset.

### E.3  20NG

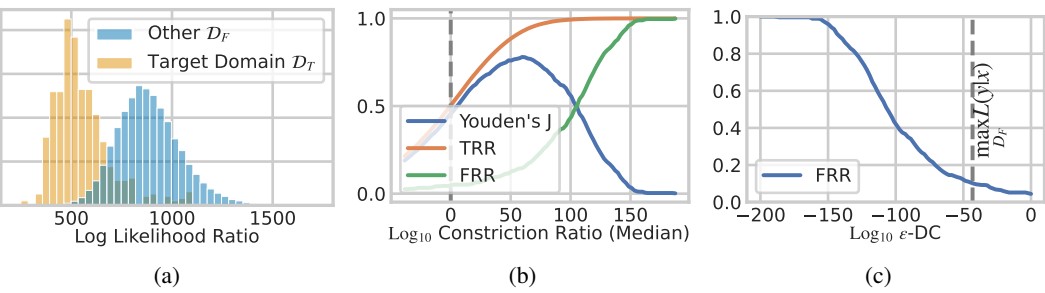

(a)  (b)  (c)

Figure 11: Figure 11a shows that log likelihood ratios are well disentangled. Figure 11b shows the trade-off between OOD and certification: The best OOD detection performance occurs with a constriction ratio of 60. Figure 11c shows the false rejection rate (FRR) required to certify at a given $\epsilon$.

### E.4  MEDICAL QA

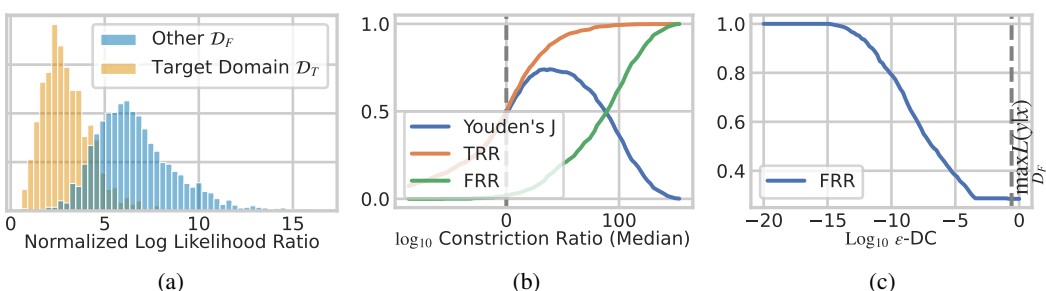

(a)  (b)  (c)

Figure 12: Figure 12a shows that log likelihood ratios are well disentangled. Figure 12b shows the trade-off between OOD and certification. Figure 12c shows the false rejection rate (FRR) required to certify at a given $\epsilon$. All results are for VALID with $T = 1$ for Medical QA.

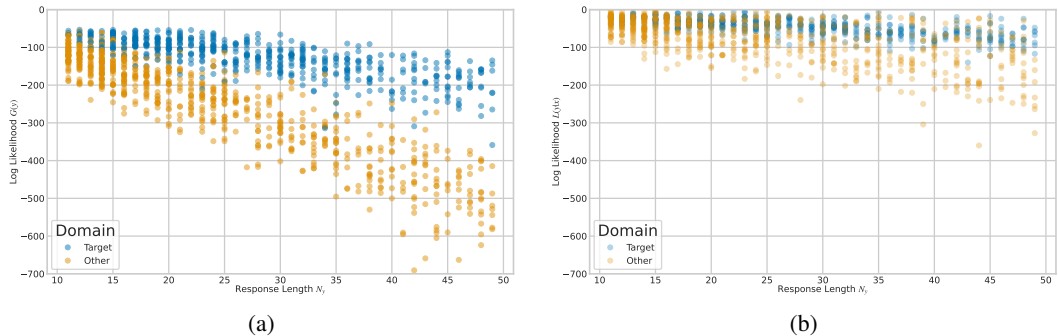

(a)  (b)

Figure 13: This figure demonstrates the gap in log likelihood between in-domain and out-of-domain samples for the guide models $G$ in Figure 13a and the LLM $L$ in Figure 13b. As the length of the response, $N_{\boldsymbol{y}}$, increases, the gap between ID ($\mathcal{D}_{\mathbb{T}}$) and OOD data ($\mathcal{D}_{\mathbb{F}}$) widens. The log-likelihood decreases roughly linearly. Thus, the guide model $G$ on the left side assigns exponential decreasing probabilities to OOD samples.

E.5 CONSTRICTION RATIOS FOR DIFFERENT FALSE REJECTION RATES

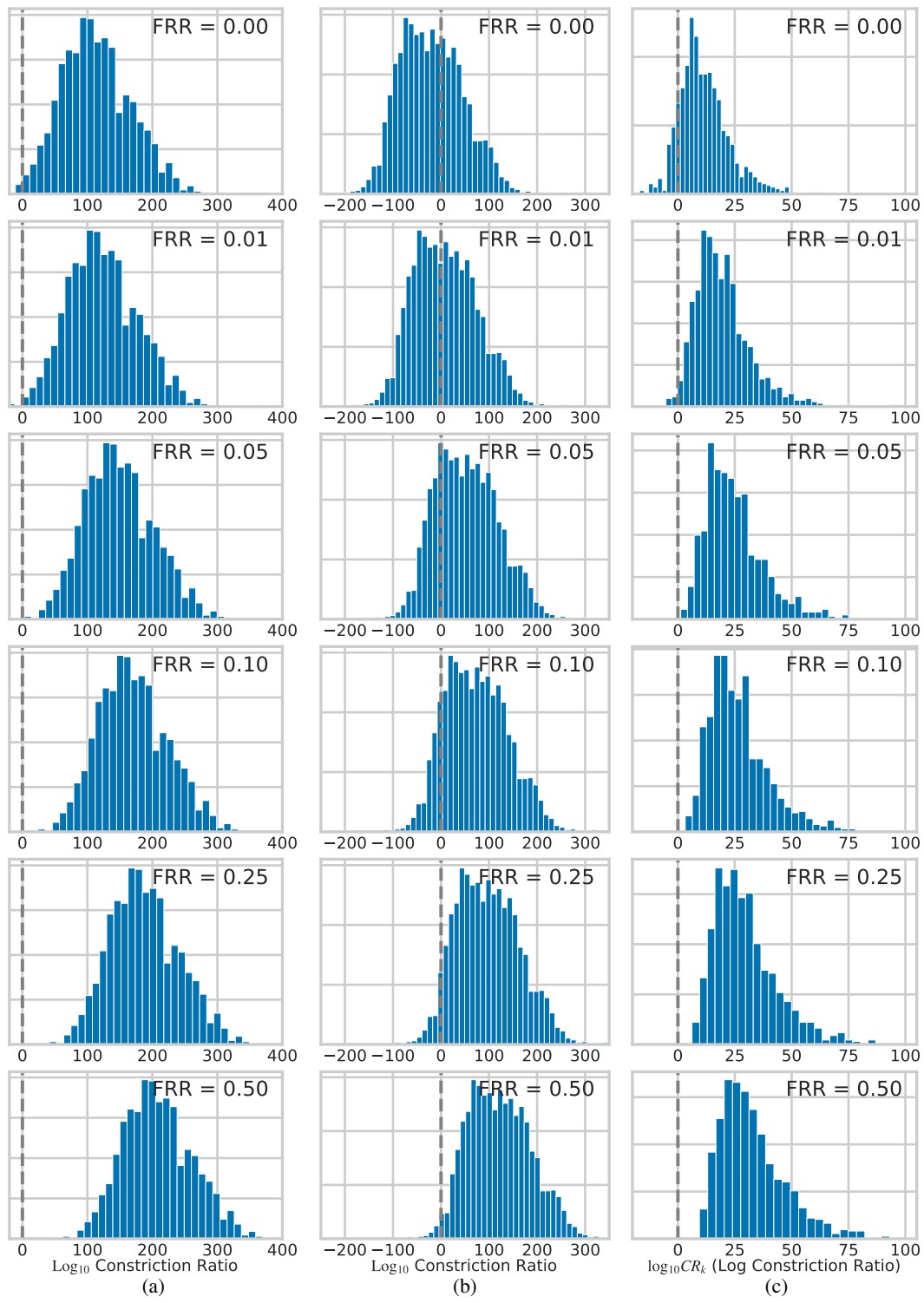

Figure 14: This figure shows the $log_{10}$ constriction ratios (CR) on OOD samples as a function of the false rejection rate (FRR) on the in-domain samples. The rejection threshold $k$ is systematically decreased from top to bottom to achieve a given FRR. We can observe the gradual improvement in constriction while increasing the FRR. (a) shows Tiny Shakespeare, (b) shows 20NG, and (c) Medical QA.

### E.6 ATOMIC CERTIFICATES - LENGTH CONTROLLED

The experimental setup for MedicalQA uses Pub-
MedQA as in-domain dataset and SQuAD as out-of-
domain dataset as described in Section 3.1 and Ap-
pendix D. The different lengths of responses in these
datasets confound our findings on the disentanglement
of the atomic certificates, $\epsilon_{\boldsymbol{y}}$-ACs between in-domain
data, $\mathcal{D}_{\mathbb{T}}$, and out-of-domain data, $\mathcal{D}_{\mathbb{F}}$. In Figure 15,
we show that sequences tend to be a lot shorter in $\mathcal{D}_{\mathbb{F}}$
than in $\mathcal{D}_{\mathbb{T}}$. As the likelihood of a response decays ex-
ponentially in the length of the responses, the responses
in the OOD set $\mathcal{D}_{\mathbb{F}}$ have relatively high likelihood that
is not attributable to the domain restriction, but rather to
the length of the response. This results in the eCDFs in
Figure 4b overlapping significantly. To show that this
is a confounding factor that is indeed worsening disen-
tanglement, we resample the data to account for length
and present results here.

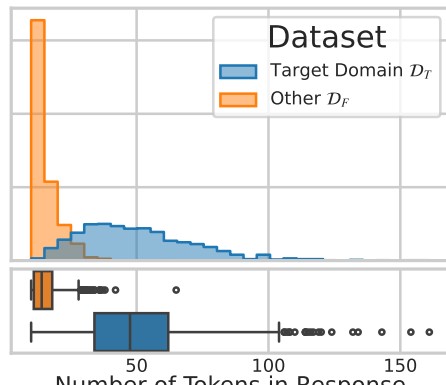

Figure 15: MedQA setup: The in-domain
dataset (PubMedQA) has longer responses
than the OOD dataset (SQuAD).

**Setup.** We resample the in-domain data, $\mathcal{D}_{\mathbb{T}}$, and
out-of-domain data, $\mathcal{D}_{\mathbb{F}}$ to have matching distribution
of response lengths. We find the target distribution us-
ing the following steps: First, we find the common support between the distribution of response
length $N_{\boldsymbol{y}}$ between $\mathcal{D}_{\mathbb{T}}$ and $\mathcal{D}_{\mathbb{F}}$, $N_{\boldsymbol{y}} \in [15, 38]$. This interval covers 67% of samples in the target
domain dataset and 58% of the OOD dataset. Second, we obtain the empirical distribution of $N_{\boldsymbol{y}}$
in the in-domain dataset, perform Laplace smoothing (Manning et al., 2008) with $\alpha = 1$ and then
further smooth the distribution using a moving average with a window length of 5. Third, we per-
form weighted sampling with replacement from $\mathcal{D}_{\mathbb{T}}$ and $\mathcal{D}_{\mathbb{F}}$ with a size of 100 times the original.
The sampling weights are computed s.t. the distribution of $N_{\boldsymbol{y}}$ matches the target distribution. We
denote these **re**sampled sets as $\mathcal{D}_{\mathbb{T}}^{RS}$ and $\mathcal{D}_{\mathbb{F}}^{RS}$.

**Results.** We find that the disentanglement of atomic
certificates, $\epsilon_{\boldsymbol{y}}$-ACs, improves greatly after eliminat-
ing the confounding factor response lengths. Figure 16
shows the empirical cumulative distribution functions
(eCDFs) for "original" datasets, $\mathcal{D}_{\mathbb{T}}$ and $\mathcal{D}_{\mathbb{F}}$ in gray
tones, as well as the results for $\mathcal{D}_{\mathbb{T}}^{RS}$ and $\mathcal{D}_{\mathbb{F}}^{RS}$. You
may observe that the distribution of ACs shifted left
for datasets representing $\mathbb{F}$ and shifted right for datasets
representing $\mathbb{T}$, effectively increasing the disentangle-
ment. This indicates that, when comparing *similar* in-
domain and out-of-domain samples, the gap in restric-
tion is larger than presented in Figure 4c. ACs on in-
domain samples are more *permissiveness* and ACs on
out-of-domain samples even more *constrictive* than it
initially appeared.

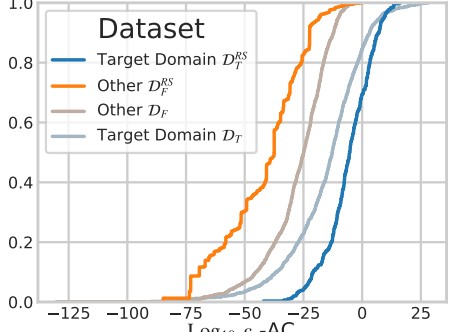

Figure 16: The eCDFs of $\epsilon_{\boldsymbol{y}}$-ACs are shown
for the original in- and out-of-domain data for
the MedQA setup in comparison to a resam-
pled dataset controlling the response length as
confounder. The gap between the permissive-
ness of in-domain samples and restrictiveness
on out-of-domain samples is greatly improved.

### E.7 ATOMIC CERTIFICATE BY LIKELIHOOD

Obtaining a tight atomic certificate for a sample $\boldsymbol{y}$ is
most important when the sample is likely proposed by
$L$. Hence, in this section we study the log constriction ratio, the tightening of our adversarial certifi-
cate over model $L$, as a function of the sample's likelihood under $L$.

We bin out-of-domain samples into 10 bins based on their log likelihood under model $L$, i.e.
$\log L(\boldsymbol{y}|\boldsymbol{x})$, and compute median, 25th and 75th percentile log constriction ratio, as well as the
median log likelihood. We present results in Figure 17 for both 20NG and TinyShakespeare. We
observe that the constriction strengthens when samples get more likely under $L$. That means, those
samples most likely to be sampled under $L$ benefit most from our atomic certificate. We consider
this to be a favorable result.

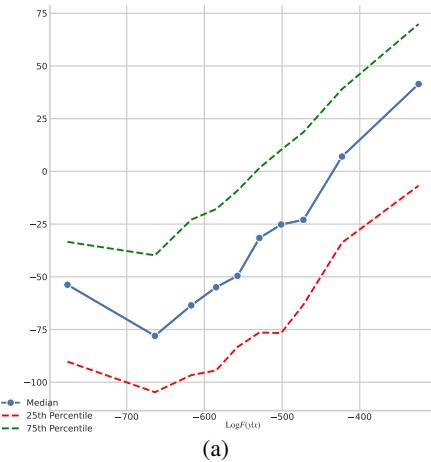 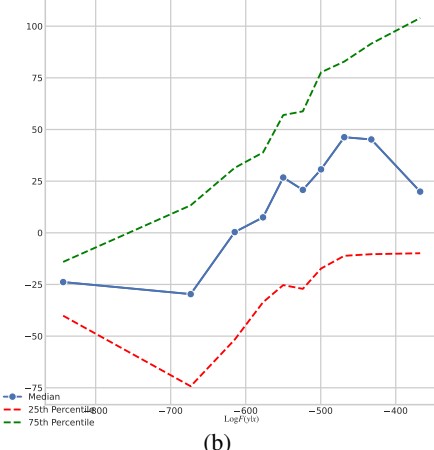

(a)                                                       (b)

Figure 17: These figures show the constriction ratio as a function of log likelihood of samples under $L$ for out-of-distribution samples. Figure 17a displays the results for 20NG and Figure 17b for TinyShakespeare. We bin all samples into 10 bins. For each bin the $x$-axis shows the median log likelihood of the sample under $L$, $\log L(\boldsymbol{y}|\boldsymbol{x})$. The $y$-axis shows the $\log_{10}$ constriction ratio (median and percentiles for each bin).

## F  REPEATED SAMPLING ($T > 1$)

In Section 3.3 we study the performance of VALID by sampling from $L$ with a single step, that is, $T = 1$. Here, we extend the analysis to $T > 1$.

**Setup.**  We adopt the MedicalQA setup as described in Appendix D.5. However, instead of employing VALID with $T = 1$, we use $T \in \{1, 2, 3, 4, 5\}$ and study the resulting $\epsilon - DC$ for combinations of $k$ (the rejection threshold of VALID). As above, for ease of presentation we use a fixed temperature of $1.0$ for $L$.

**Results.**  We find that increasing $T$ significantly reduces false rejection rates (FRR) while only marginally increasing the $\epsilon$-DC (domain certificate). We present findings for the FRR in Figure 18a and for $\epsilon$-DC in Figure 18b. The minor increase in $\epsilon$ due to increasing $T$ should not come as a surprise as we recall the formula for the upper bound: $2^{kN_{\boldsymbol{y}}}TG(y)$ (see (4)). Even $T = 10$ increases the upper bound $\epsilon_y$ by only one order of magnitude. On the other hand, the gains in in-domain performance are marked. In Figure 18a, we can observe the FRR is roughly halved for $T = 5$ and $k > 2$, greatly improving the refusal behavior of the model on in-domain samples. Finally, we note that the temperature of $L$, $t_L$, is a confounding factor. For $t_L \to 0$, we would perform (nearly) deterministic sampling of $\boldsymbol{y}|\boldsymbol{x}$ and therefore $T > 1$ would not have any benefit.

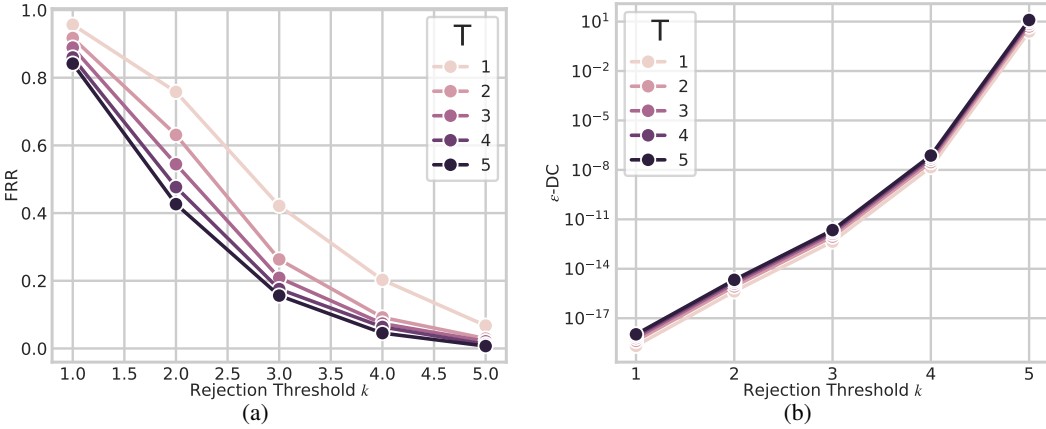

Figure 18: False Rejection Rate (FRR) (a) and $\epsilon - DC$ of the Domain certificate of VALID (b) plotted for a range of different values of $T$ and $k$.

## G   ABLATION

### G.1   COMPARING $M$ TO $G$

Our goal is to provide a guarantee on a generalist model assuming that such a model outperforms custom, small solutions that are inherently safer due to their domain specific training. We test this empirically by examining the gap in performance between the generalist model $L$, a small in-domain model. As $G$ is trained marginally on $\boldsymbol{y}$, it is not able to perform any task. Hence, we exactly replicate the training procedure of $G$ and train a model on the entire sequence, $G'(\boldsymbol{x}, \boldsymbol{y})$. We utilize the CharTask dataset as described above and study the accuracy of each model in generating valid sequences: A valid sequence is one that starts with Q, is followed by a random sequence of characters (e.g. 5 3), followed by four unique task tokens (e.g. S A E R) defining a task, which is then performed (e.g. 3 5). The sequence is expected to terminate there. If *any* of these are violated, the generated sequence is scored as invalid. We perform inference on 1000 prompts from the target domain test dataset prompting the model with various lengths of prompts. In Table 3, we present the results: The accuracy of generating such sequences of $L$ lies significantly above that of $G$ (difference of approx 30%). This shows that $G$ is effective in restricting the domain while performing considerably worse than $L$. Hence, our method combines the best of both models: The safety of $G$ with the performance of $L$.

| Prompt Length | G | L |
|---|---|---|
| 1 | 60.45 | 91.21 |
| 5 | 60.25 | 92.68 |
| 10 | 66.89 | 91.11 |

Table 3: Accuracy scores for CharTask generation dataset.

### G.2   BENEFIT OF LARGER GUIDE MODELS

In this Appendix, we study the influence of the size of $G$ on the VALID results. In particular, we ask whether VALID benefits from smaller or larger models.

**Setup.**   We turn to our MedicalQA setup as described in Section 3.1 and Appendix D.5. With the same methodology, we fit two more models for $G$. $G_{XS}$ follows a GPT-2 architecture with 6 layers, 6 heads and 192 embedding dimension resulting in 27.49M parameters. $G_S$ follows a GPT-2 architecture with 6 layers, 6 heads and 384 embedding dimensions resulting in 60.29M parameters. To recap, the $G$ model as used above uses 12 layers, 12 heads and 768 embedding dimension resulting in 184M parameters. We then compare the three models on samples generated by $L$ following Section 3.3.

**Results.**   We find that larger models tend to perform better, however, the evidence is not strong. First, we study the rejection threshold $k$ per model. As described in (4) in Theorem 1, VALID's upper bounds gets tighter with smaller $k$. Hence, in Figure 19a we plot $k$ values achieving a given false rejection rate (FRR) for each model. We observe that larger models enable smaller $k$ at the

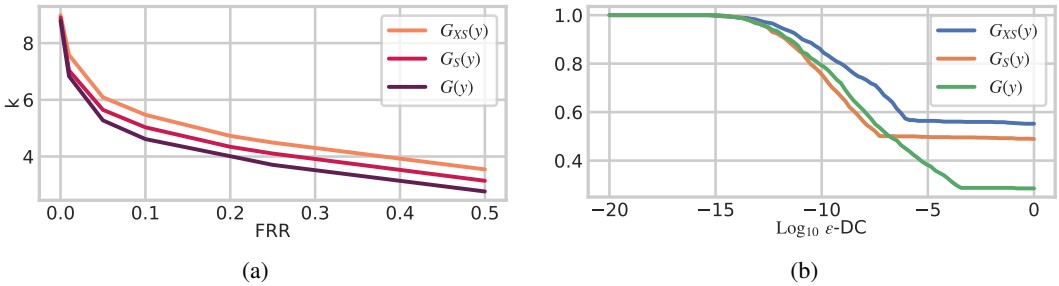

Figure 19: These Figures demonstrate differences in the behavior of VALID for different sizes of guide models $G$. Figure 19a shows that larger models allow for lower $k$ and hence lower bounds at the same False Rejection Rate (FRR). Figure 19b shows the FRR ($y$-axis) for a given $\epsilon$-DC for guide models of different sizes.

same FRR. This indicates that the trade-off in $k$ between certification and OOD detection is more favorable under larger models. This should not come as a surprise, as larger models tend to achieve better perplexity (i.e. lower loss) on in-domain data.

Next, we study the constriction ratios of the Atomic Certificates (AC) and present results for different sizes of $G$ as shown in Table 4. For each model, we provide the the 10th percentile, median and 90th percentile. You may observe that $G_{XS}(y)$ consistently provides constriction ratios that are are often around 10 orders of magnitudes worse than $G_S(y)$ and $G(y)$. Interestingly, $G_S(y)$ yields better ratios than $G(y)$. However, the difference is smaller. We speculate that the limited amount of ID training data means we do not see benefits for increasing the size of $G$ beyond a point, as it begins to overfit without increasing regularization.

Finally, we study the Domain Certificates (DC) for each model. For this we replicate Figure 12c and present Figure 19b showing the false rejection rate (FRR) given an $\epsilon$-DC for the three models. We may observe that the lower bound to the FRR significantly increases as the models become smaller. The evidence here suggests that larger guide models yield better domain certificates.

In conclusion, the evidence points to larger models working better for an application like MedQA. The evidence uniformly shows that a model as small as $G_{XS}(y)$ does perform significantly worse than larger models.

| FRR | Log$_{10}$ Constriction Ratio (10% / Median / 90%) | | |
|---|---|---|---|
| | $G_{XS}(y)$ | $G_S(y)$ | $G(y)$ |
| **0%** | -427 / -45 / 12 | -408 / -41 / 12 | -449 / -54 / 6 |
| **1%** | -246 / -14 / 42 | -176 / -3 / 79 | -198 / -10 / 43 |
| **5%** | -74 / 12 / 141 | -42 / 21 / 195 | -42 / 18 / 162 |
| **10%** | -29 / 24 / 202 | -11 / 35 / 257 | -8 / 33 / 229 |
| **20%** | -3 / 43 / 281 | 1 / 57 / 337 | 3 / 50 / 302 |
| **25%** | 0 / 50 / 308 | 5 / 63 / 364 | 7 / 60 / 345 |
| **50%** | 11 / 81 / 430 | 13 / 96 / 497 | 15 / 89 / 477 |

Table 4: Constriction Ratios for MedicalQA for three models of different sizes. The smallest model yields significantly worse (lower) constriction ratios.

## H   BENCHMARKING

In this section, we provide a comprehensive description of the PubMedQA experimental setup presented in Section 3.4, present additional benchmarking results, and extend our evaluation framework to the MMLU benchmark (Hendrycks et al., 2021).

### H.1   PUBMEDQA

**Setup.**    The PubMedQA benchmark (Jin et al., 2019) comprises 1000 items. Each item contains background information (context), a multiple-choice question (answerable by yes/no/maybe), a long-text answer, and a ground truth label (yes/no/maybe). As illustrated in Figure 6, we evaluate the model through two streams: "item correctness" and "response acceptance". In both streams, we prompt the model with the context and question. In the "item correctness" stream, the model is provided with all multiple-choice tokens, and the maximum likelihood answer is selected and evaluated for correctness. In the "response acceptance" stream, we present the long-text answer as a response and determine if model $M$ abstains at a given domain certificate of $\epsilon$. An item is considered correct at $\epsilon$ if and only if the response is accepted and the model scores correctly. We use the reasoning-required variant of the PubMedQA benchmark (for further details, see Jin et al. (2019)).

**Results.**   Extending our analysis of the PubMedQA benchmark presented in Section 3.4, we examined the relationship between PubMedQA performance scores and median constriction ratios. As illustrated in Figure 20a, our findings demonstrate that the model can achieve a $\log_{10}$ constriction ratio of 20 on samples in $\mathcal{D}_\mathbb{F}$ while maintaining robust PubMedQA performance. Specifically, at a performance threshold of 70% accuracy, we observed a $\log_{10} CR_k$ value of 21.6, which effectively

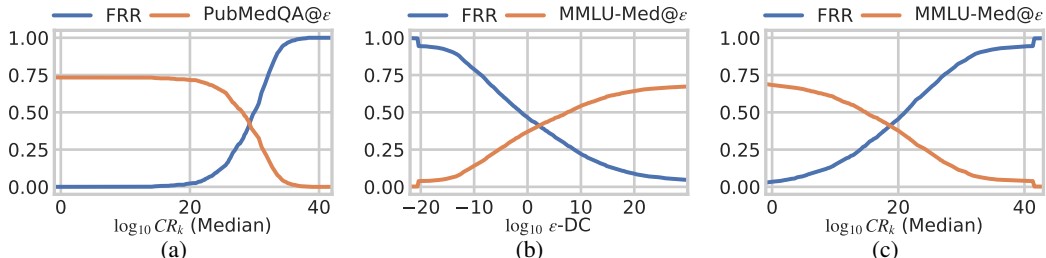

Figure 20: Evaluation of domain-certified models through standardized benchmarking. Figure 20a illustrates the association between $\log$ constriction ratios and the PubMedQA@$\epsilon$ benchmark scores across models with varying $\epsilon$-DC certifications. Figure 20b presents the MMLU@$\epsilon$ metric evaluated at different certification thresholds $\epsilon$. (c) Figure 20c shows the relationship between $\log$ constriction ratios and corresponding MMLU@$\epsilon$ scores across multiple certification levels.

constrains out-of-domain samples to probabilities at least $1 \times 10^{-21}$ times lower than the likelihood of samples in distribution $L$. This indicates a strong capacity for domain constriction while preserving task performance.

## H.2 MMLU-MED

In this section, we extend the benchmarking of our certified model $M$ for medical question answering to the MMLU benchmark (Hendrycks et al., 2021). To that end, we follow the same methodology as above for the PubMedQA benchmark. In an earlier version of this work, we reported MMLU results that were erroneous, which we correct here.

**Setup.** MMLU comprises thousands of questions spanning various domains of general and professional factual knowledge. As our model $M$ is deployed for medical questions, we focus on a subset of MMLU categories that fall within our domain $\mathbb{T}$. We specify the selected categories in Table 2 and designate this remaining benchmark as MMLU-Med.

MMLU's standard format provides $n$-shot examples with four possible answers (A through D) followed by a question in the same format. The model is then prompted to select the correct response. However, this setup does not reflect a realistic user-system interaction. Therefore, similar to PubMedQA, we introduce the MMLU-Med@$\epsilon$ metric, which separates the evaluation into two streams: (1) standard assessment of model $L$ on MMLU-Med to determine correctness, and (2) testing whether the correct question-answer pair is rejected by our algorithm. The process is summarized in Figure 21. We score an item as correct when the model scores correctly while maintaining its $\epsilon - DC$ on the realistic question-answer pair.

**Results.** Our evaluation yields mixed evidence regarding the model's performance on MMLU-Med. Following the same analysis as for PubMedQA in Section 3.4, we present the MMLU-Med@$\epsilon$ metric in Figure 20b. As shown, MMLU-Med@1 = 37.1%, that is, the model retains 37.1% accuracy when certified at $\epsilon = 1$, or $\log_{10} \epsilon = 0$. The $10^{-10}$-DC model achieves a score of 14.1%. In addition, to the domain certificates, we study the median constriction of our model in relation to

| MMLU-Med @ $\epsilon$: Set $k$ s.t. $M$ is $\epsilon - DC$ on $\mathscr{D}_{\mathbb{F}}$ | |
|---|---|
| Question Correctness | Accepting Response |
| Which of the following is true of Graves Disease of the thyroid? 
 ✅ A: It is a cause of ophthalmoplegia 
 ❌ B: It causes a large multi-nodular goitre 
 ❌ C: It is commoner in males than females 
 ❌ D: In the past, Grave's disease sometimes caused 'Derbyshire Neck' 
 Answer: {A,B,C,D} | Question: Which of the following is true of Graves Disease of the thyroid? Answer: It is a cause of ophthalmoplegia 
 $\log L(\mathbf{y} \mid \mathbf{x})/G(\mathbf{y}) \leq kN_y$ ✅ |
| Correct Answer & Answer Accepted. Question Score: ✅ | |

Figure 21: The MMLU@$\epsilon$ benchmark assesses MMLU performance while satisfying $\epsilon$-DC certificate. The correctness is scored as commonly done for MMLU (left). The correct question answer pair is checked for acceptance / rejection by $M$. Only if a sample is accepted and correct, the question is scored positively. For questions not ending in "?", the sentence is concatenated without keywords.

its certified performance. The evidence provided in Figure 20c indicates that a median constriction ratio of $1 \times 10^{-5}$ is achieved on out-of-domain samples together with a score of $65\%$ on the MMLU-Med@$\epsilon$ benchmark. Further, a median constriction of $1 \times 10^{-20}$ is achieved with an MMLU-Med@$\epsilon$ score of $37\%$.

These results are considerably weaker than the strong results presented above for PubMedQA raising the question as to why this is. In Figure 22, we investigate the domain shift between PubMedQA and MMLU. In particular, Figure 22a shows the distribution of log likelihoods of in-domain samples ($\mathcal{D}_{\mathbb{T}}$), out-of-domain samples ($\mathcal{D}_{\mathbb{F}}$) and MMLU samples under guide model $G$ and Figure 22b shows the log likelihood ratios for the same samples. We observe a considerable overlap between the distributions of MMLU-Med and PubMedQA samples. However, the distribution of MMLU-Med has a long-tail into the distribution of $\mathcal{D}_{\mathbb{F}}$. This explains our results quite well. On the one hand, the large overlapping mass of MMLU-Med and PubMedQA explains why $M$ accepts a wide range of MMLU-Med responses while significantly constricting the model on $\mathcal{D}_{\mathbb{F}}$. On the other hand, the long tail of the distribution of MMLU scores into the distribution of $\mathcal{D}_{\mathbb{F}}$ indicates a range of MMLU-Med questions will be rejected unless the certificates become vacuous, making it hard challenging high MMLU-Med@$\epsilon$ performance. We believe that training $G$ on MMLU-style QA pairs would significantly improve results but leave this as a future direction.

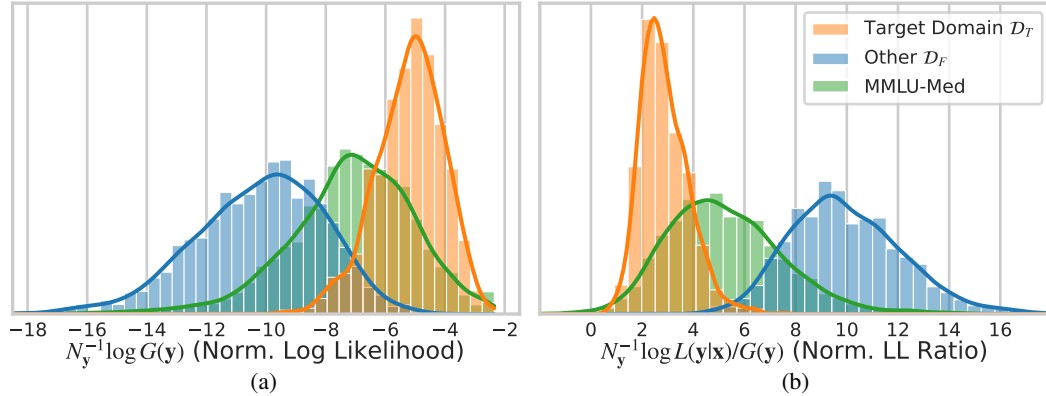

Figure 22: Comparison of likelihood of three datasets under model $G$, showing a MMLU-Med exhibits domain shift relative to PubMedQA. Figure 22a indicates likelihood of MMLU-Med lies in-between the in-domain data $\mathcal{D}_{\mathbb{T}}$ (PubMedQA) and out-of-domain data $\mathcal{D}_{\mathbb{F}}$. Figure 22b shows the normalized log likelihood ratio used in VALID lead to frequent rejections due to domain shift.

