# OpenReview forum: "Shh, don't say that! Domain Certification in LLMs"
_ICLR.cc/2025/Conference — ICLR 2025 Poster_

### Official Review · Reviewer_XBJU · 2024-11-01

**Soundness:** 4
**Presentation:** 4
**Contribution:** 3
**Rating:** 8
**Confidence:** 4

**Summary:**

The authors propose a method to assess and mitigate the risk of Large Language Models (LLMs) deployed in particular tasks answering out-of-domain questions, a concept/framework which they formalize as domain certification. The proposed method, VALID, provides adversarial bounds as a certificate. Finally, this method is evaluated over 3 datasets.

**Strengths:**

The paper is interesting, well motivated and tackles a relevant problem.

The problem and concepts are well-defined. Domain certification is a interesting concept, and VALID's design appears well justified.

Despite the lack of discussion and comparison on existing guardrails this method benefits from providing some type of mathematical guarantees, which is a desirable property given recent legislation (particularly in the EU), as the authors mention.

**Weaknesses:**

The literature review should be improved. There is little to no discussion on related methods/tasks, and existing approaches to tackle this issue (if any). For example, a greater discussion on methods regarding LLM guardrails, and how they compare to your approach.

Domain certification always requires the definition of F, which if I understood correctly, is used for the definition of domain certification and the experimental procedure, since VALID was motivated mainly by the "certification through divergence" approach. Adding some general recommendations on the definition of F in general terms, or for applied scenarios, would be interesting (e.g., I could find vast amounts of OOD data in benchmark datasets for a tax advisory LLM, but is there an efficient way to select a representative F?)

It would be interesting to see how different definitions of the model G affect the quality of this method.

The experimental results lack a comparison to any benchmark or baseline method. Literature on LLM guardrails could be used as a way to compare the effectiveness of this approach. In my opinion, the main and most important weakness of this paper is the experimental setup, followed by the literature review, both of which should be substantially improved.

I am accepting this paper under the assumption these weaknesses will be addressed, and am willing to improve the overall score at a later stage, depending on the quality of the revised version.

**Questions:**

- How does this method compare to setting up restrictive (already existing) guardrail methods specific to a domain/task?
- If the LLM is being deployed for a specific task (such as the tax report example used in throughout the paper), how easy is it to set up VALID? Do you use a finetuned version of L to form G? Do you use a smaller finetuned LLM? Is it trained from scratch? Is it a simple discriminator? This is something that may be clarified once source code is available, but in any case it would have been great if at least an anonymized repository was provided.
- What is contained in F′ ∩ T′? Seems unclear to me. Only semantically incoherent sequences?
- VALID requires a secondary language model G, trained on in-domain data. In that case, since domain data is necessary anyway, wouldn't it be simpler to just quantify similarities between an output and the in-domain corpus within an embedding space, sparing the need for secondary model trained with in-domain data?
- Around line 150, you mention "The deployer might perform an a priori risk assessment and determine that they can tolerate the consequences of one out-of-domain response from a set D_F per year." But given your method depends on F for domain certification, and VALID depends on G, are there any guarantees that only one out-of-domain response will be produced per year? Seems like a very bold claim.

**Details Of Ethics Concerns:**

No concerns

---

> ### Author Response · Authors · 2024-11-25
> **Rebuttal 1/3**
>
> We thank the reviewer for their detailed comments. We are delighted they find our work is "well motivated and tackles a relevant problem". We appreciate the reviewer stating that our methods are "well justified".
>
> > W1: The literature review should be improved. There is little to no discussion on related methods/tasks, and existing approaches to tackle this issue (if any). For example, a greater discussion on methods regarding LLM guardrails, and how they compare to your approach.
>
> We believe our works sits at the intersection of a number of fields such as a) (adversarial) LLM guardrails b) other forms of certification in NLP (e.g. text classification) and formal verification c) OOD detection in NLP. In our introduction (L60-L62), we mostly focus on papers that are close to the intersection of these topics rather than providing a full reference list for each domain. Following the reviewer's suggestions, we added a dedicated "Related Works" section covering these areas individually in more depth. This will help the reader gain some background information.
>
> We want to stress that we are the first to consider the problem of domain certification, so there is nothing that is directly comparable. Specifically, there are some key differences between domain restriction and general LLM guardrail for alignment, in that VALID produces certificates that hold for all inputs. To the best of knowledge no other guardrails come with such guarantee.
>
> > W2: Domain certification always requires the definition of F, which if I understood correctly, is used for the definition of domain certification and the experimental procedure, since VALID was motivated mainly by the "certification through divergence" approach. Adding some general recommendations on the definition of F in general terms, or for applied scenarios, would be interesting (e.g., I could find vast amounts of OOD data in benchmark datasets for a tax advisory LLM, but is there an efficient way to select a representative F?)
>
> This is a really interesting question. In order for the guarantees to be meaningful, $D_F$ must be selected to be representative of $F$, and the more time spent crafting $D_F$ the more useful the guarantee. We note there is a strong precedent for using finite samples for evaluation in most ML frameworks, including certifcation. For example, adversarial *certified* accuracy in the image domain gives no guarantees on generalisation to elements outside the finite test set.
>
> When deploying VALID as a service, one would ask the client to use domain knowledge to help characterise a threat model and hence $F$, and $D_F$. For example selecting $D_F$ to contain the set of particularly harmful outputs from a Public Relations point of view. This base set of harmful examples could then be extended during red teaming or inflated in-size using paraphrasing and other-similar automatic methods. For certifying against misuse it would be much harder to get good coverage of all topics considered out of domain, but a representative sample of likely misuse cases should be sufficient to determine if VALID protects against them. We would recommend selecting to $D_F$ to contain common general chat bot queries or a large amount of easily available off topic data sets, such as In Section 3 of the paper.
>
>
> > W3: It would be interesting to see how different definitions of the model G affect the quality of this method.
>
> We thank the reviewer for this suggestion. We have added Appendix E.2 in which we run an ablation on the size of model $G$. We find evidence that small $G$ might not allow the model to generalise well enough in-domain. Hence, VALID tends to perform better with larger models, if they don't overfit.
>
>
> > W4a: The experimental results lack a comparison to any benchmark or baseline method. Literature on LLM guardrails could be used as a way to compare the effectiveness of this approach.
>
> We are the *first* to introduce this kind of certificate. While we could compare the OOD detection ability of VALID against other methods, we would not be able to compare certificates. In other words, to the best of our knowledge there are no other methods that give similar guarantees for all inputs. This would limit the usefulness of these comparison.
>
> The constriction ratios (see Eq 5 and Table 1) could be seen as a very crude comparison against approximate certificates on $L$. We hope this metric will be adopted for further comparisons.
>
> A lot of guardrails focus on notion of safety for the end user, while having applications in safety is broader. "Help me with my math homework" is not something LLM guardrails are used for, however, might be relevant for a deployer trying to prevent misuse of their resources.

---

> > ### Author Response · Authors · 2024-11-25
> > **Rebuttal 2/3**
> >
> > > W4b: In my opinion, the main and most important weakness of this paper is the experimental setup
> >
> > If possible, it would be great if the reviewer could clarify to which aspects of the experimental setup they are referring. We would like to point out that we have added further experiments following the suggestions of reviewers, e.g. the ablation on the size of $G$ in Appendix E.2.
> >
> > > W5: literature review, [...] should be substantially improved.
> >
> > We refer the reviewer to our response to W1.
> >
> > > Q1: How does this method compare to setting up restrictive (already existing) guardrail methods specific to a domain/task?
> >
> > The main difference is the kind of guarantee you get. For ease of discussion consider a single unwanted behaviour. When assessing already existing guardrail methods a standard metric is Average Attack Success Rate; [1,2], this is an emprical measurement over adversarial samples $x$ on how many generate the harmful behaviour, typically judged by another LLM. VALID gives a certificate evaluated over a representative set of ODD outputs $\mathcal{D}_{\mathbb{F}}$ (which could be generated by and LLM), against *any* inputs. We would love to hear the reviewer's suggestion into establishing a comparison across those two very different metrics.
> >
> > [1] Perez E, Huang S, Song F, Cai T, Ring R, Aslanides J, Glaese A, McAleese N, Irving G. Red teaming language models with language models. arXiv preprint arXiv:2202.03286. 2022 Feb 7.
> >
> > [2] Zou A, Wang Z, Carlini N, Nasr M, Kolter JZ, Fredrikson M. Universal and transferable adversarial attacks on aligned language models. arXiv preprint arXiv:2307.15043. 2023 Jul 27.
> >
> > > Q2: If the LLM is being deployed for a specific task (such as the tax report example used in throughout the paper), how easy is it to set up VALID? Do you use a finetuned version of L to form G? Do you use a smaller finetuned LLM? Is it trained from scratch? Is it a simple discriminator? This is something that may be clarified once source code is available, but in any case it would have been great if at least an anonymized repository was provided.
> >
> > This is an excellent question. First, we would like to note that no matter the choice of $G$ the certificate will hold either way. The influence of $G$ is on the tightness of the certificate. We train $G$ from scratch in all of our experiments using very much the standard parameters recommended by huggingface and an absolutely standard setup vis-a-vis data processing. After anonymous peer review we will share the our $G$ models on Huggingface hub.
> >
> > We believe the fact that $G$ is trained exclusively on in-domain data is the crux for VALID to distinguish well between in-domain and out-of-domain responses (see Figure 11 in the revised paper comparing the likelihood $G$ puts on ID and OOD data). This can be a custom model trained from scratch or fine-tuned from a larger (public) model that is exclusively trained on in-domain data.
> >
> > > Q3: What is contained in F′ ∩ T′? Seems unclear to me. Only semantically incoherent sequences?
> >
> > We thank the reviewer for this great question.
> >
> > The short answer is yes. $\mathbb{F}$ is the set of all undesirable strings. If misuse is a concern, such as users requesting resources to do maths homework using a government tax chat-bot, then $\mathbb{F}$ will contain all "useful" off topic content. In this case $\mathbb{F}′ \cap \mathbb{T}′$ will just contain incoherent sequences, with no perceived use.
> >
> > The long answer is it depends on what you want to certify against in your application. If a deployer only cares about public reputation (PR) damage $F$ would only contain outputs that they believe would cause PR damage, and $\mathbb{F}′ ∩ \mathbb{T}′$ would contain off-topic content that should not cause PR damage. Finally, it is up to the practitioner to decide what is in $\mathbb{F}$ and what isn't. Is it a priority to protect against an adversary eliciting the sequence "The sky is blue. " x 100?
> >
> > > Q4: VALID requires a secondary language model G, trained on in-domain data. In that case, since domain data is necessary anyway, wouldn't it be simpler to just quantify similarities between an output and the in-domain corpus within an embedding space, sparing the need for secondary model trained with in-domain data?
> >
> > Could the reviewer elaborate a bit here? $G$ does not have to be a LLM, other systems assigning high likelihood to ID samples and low likelihood to OOD samples could be used. We note if the in-domain corpus is large, storing it directly would likely be infeasible and thus it would likely need to be compress via LLM anyways, providing in-domain generalization capabilities. This is what $G$ achieves.
> >
> > We believe the barriers on $G$ are quite low, so we are not too concerned with the requirement for $G$, but yes if we can get an equally performant certificate without $G$ (in terms of certificate tightness, false rejection rates and computational expense), that would be preferred. We leave that for future work.

---

> ### Author Response · Authors · 2024-11-25
> **Rebuttal 3/3**
>
> > Q5: Around line 150, you mention "The deployer might perform an a priori risk assessment and determine that they can tolerate the consequences of one out-of-domain response from a set D_F per year." But given your method depends on F for domain certification, and VALID depends on G, are there any guarantees that only one out-of-domain response will be produced per year? Seems like a very bold claim.
>
> This example is to exemplify Definition 2 (the Domain-Certificate). If one has access to $\epsilon$ in Definition 2 for a model $M$ and a set $F$, then the interpretation of that is a worse case bound on the expected number of generations from $M$ before a violation in $F$ takes place. However, in practice $F$ is a theoretical construct which cannot be enumerated. Thus, we have the relaxation where $G$ captures the in-domain, indirectly capturing $F$, and we have an algorithmic approach VALID guaranteeing epsilon by construction. Thus, the interpretation here would be we need X number of years at generations of this rate per day, before L generates something deemed as out of domain with respect to the generator $G$.Moreover, How well a certificate on $D_F$ generalises to $F$ more generally depends on how well $D_F$ is chosen. This is a common issue with machine learning evaluation that one must pick a finite subset of examples to evaluate on and picking a poorly representative set of the distribution of interest leads to a poor estimation of how the system will perform in the wild.

---

> > ### Comment · Reviewer_XBJU · 2024-12-02
> >
> > (apologies for the delay)
> >
> > I appreciate the authors' responses to the comments made. After reading through the rebuttals and the remaining reviews/discussions, I have raised my score to reflect the answers provided and changes in the paper made by the authors.

---

> ### Author Response · Authors · 2024-12-03
>
> We thank the reviewer for their continued commitment and the efforts to read our rebuttals. We are glad that we could address the reviewer's concerns and thank the reviewer for raising their score. If there are any last minute questions, we will gladly respond.

---

### Official Review · Reviewer_8eQm · 2024-11-04

**Soundness:** 3
**Presentation:** 3
**Contribution:** 3
**Rating:** 5
**Confidence:** 4

**Summary:**

This work studies the problem of adversarial certificates for LLMs in domain specific deployment contexts. They define the problem of "domain certification" and derive a test time algorithm VALID based on a small guide/reference model, and a rejection sampling criterion, that provides such domain certificates. They quantify levels of tightness and permissiveness of their certificates using real data on out of domain and in domain samples respectively to argue for the potential utility their method in real deployments.

**Strengths:**

1. Motivate the difficulty and real world implications of certifiable test time behavior for model LLM systems.
2. Present a simple and clear algorithm to achieve their definition of domain certification.
3. Identify a clever way to harness the limited abilities of very small, cheap to train, domain specific language models to implement test time rejection sampling favoring a constrained distribution of responses.

**Weaknesses:**

1. Work under very limiting constraints of a fixed set of bad strings F. This is a practical assumption, but pervasive limitation for this work and of any other attempts to characterize the output spaces of generative models with large vocabularies, and variable size outputs. See lines 132-137 for the authors' own description of this required narrowing of scope. This always leaves room for adversaries to circumvent certificates via finding inputs in the complement of the finite sample of F chosen, but still in T' such that they are falsely permitted by the system.

2. It's not immediately clear how good the D_T to D_F ratios are in Table 1. Additionally, there seems to be a correlation between success/separation and the domain similarity, which is expected? Maybe the authors can discuss this further potentially highlighting a limitation of the approach based on how well separated good and bad behaviors are in a given domain.

3. The generative results are weaker than the fixed sample set results, summarized lets say as weaker Youden's J for the same domain setting, and since this is the more realistic evaluation, it is important to caution that this approach isn't quite deployable at the moment. (On that note, lines 145-152 are more appropriate moved to the final paragraphs as a discussion section/future work/applications blurb)

Together, the problem formulation and empirical results leave the work in a somewhat middling position in terms of contribution to the field, but depending on the potential discussion surrounding questions below, this might be improved.

**Questions:**

Primary question (multi-part, single theme):

Initially, I was going to comment that training guide LLM, G, on only "good strings" in T, will not necessarily give you a model that only assigns high likelihood to strings in T, and therefore there will likely exist some strings for which both L and G assign high likelihood but are in fact in T'... however, after reaching the Section 3.1 experimental setup detail that guide models G were to be parametrized by relatively tiny gpt2 style models, it became clear how this might work in practice.

1. As a preliminary ask, "Such a certificate with respect to G can be useful: As G is only trained on samples in DT ⊂ T, a dataset of domain T, it assigns exponentially small likelihood to samples that are in F."
Is there an analytic or empirical argument that the authors can provide for this statement? It seems like both the benign and adversarial robustness of the certificate hinge on this assumption.

Now, the section of future work describes a desire to explore using stronger models for G, which suggests that the authors don't necessarily see the _weakness of G_, eg. its inability to predict next tokens in any context beyond the small corpus on which it was fit, and maybe poorly even within that domain, as the actual crux of the method that _makes it work_.

2. Was any ablation done about the model size for G and or the corpus size for training it? My hypothesis is that the stronger G becomes (more parameters and/or data), the wider set of strings it is able to model and therefor the closer L(y|x) and G(y) become, eventually depleting all the power of the algorithm.

3. Overall, can the authors elaborate on any "behind the scenes" details that aren't in the draft or appendix on how G's role in the certificate was developed during the research and why the particular experimental choices (tiny gpt2's) for it were made?

Whatever the answers are to this series of questions are, in my opinion, necessary additions to the draft to improve its clarity and soundness, as well as the ability for future work to build upon these results. Currently, my takeaway centers the identification of a weak G to compute an odds ratio against L as a way to perform rejection sampling for safety and alignment, and thus pitching the results a little more in this direction potentially improves the relevance and generalizability of the work (since the particular certificates/and algorithm might be limited in immediate applicability).

Some relevant works that abstractly leverage the same odds ratio paradigm, though for different purposes, are Contrastive Decoding by Li et al. (2210.15097) which moderates policy L via a weaker policy L' at decoding time to improve output quality, and Direct Preference Optimization by Rafailov et al. (2305.18290) as a way to train a policy L to prefer samples in T rather than T'.

Minor:

4. L257, does "OOD" mean "F"? To match L 269 referring to other question domains as "F". Generally, maybe prefer one notation or the other to refer to the "bad" set everywhere.

---

> ### Author Response · Authors · 2024-11-25
> **Rebuttal 1/3**
>
> We thank the reviewer for their detailed comments. We are glad they believe our work "Identifies a clever way to harness the limited abilities of very small, cheap to train, domain specific language models".
>
> > Work under very limiting constraints of a fixed set of bad strings F. This is a practical assumption, but pervasive limitation for this work and of any other attempts to characterise the output spaces of generative models with large vocabularies, and variable size outputs. See lines 132-137 for the authors' own description of this required narrowing of scope. This always leaves room for adversaries to circumvent certificates via finding inputs in the complement of the finite sample of F chosen, but still in T' such that they are falsely permitted by the system.
>
> Indeed, the reviewer is correct this is a weakness of our work. Ideally one would have access to the set of all unwanted responses $F$, however in practice $F$ is a theoretical construct which most likely cannot be enumerated. Thus instead we make use of a finite subset, $D_{F}$ on which certificates are evaluated. Moreover, this is the reason VALID focuses on only producing ID responses rather than not generating OOD ones. I believe the reviewer's critic is how do we know that the DC-certificate generalises to elements in $F$ from $D_{F}$?
>
> In order for the guarantees to be meaningful $D_{F}$ must be selected to be representative of $F$, and the more time spend crafting $D_{F}$ the more useful the guarantee. But we do not see this as a uncommon weakness as it is true of most data set selection strategies. We note there is a strong precedent for using finite samples for evaluation in most ML frameworks, including certification. For example (Adversarial) Robust Accuracy in the image domain gives no guarantees on generalisation to elements outside the finite test set. When deploying VALID, one would use domain knowledge and a threat model to characterise $F$, and thus $D_{F}$. For example selecting $D_{F}$ to contain the set of particularly harmful outputs from a Public Relations point of view. This base set of harmful examples could then be extended during red teaming or inflated in-size using paraphrasing and other-similar automatic methods. For certifying against misuse it would be harder to get good coverage of all topics considered out of domain, but a representative sample of likely misuse cases should be sufficient to determine if VALID protects against them. We would recommend selecting to $D_{F}$ to contain common general chat bot queries or a large amount of easily available off-topic data sets, such as we do in Section 3 of the paper.
>
> > It's not immediately clear how good the D_T to D_F ratios are in Table 1. Additionally, there seems to be a correlation between success/separation and the domain similarity, which is expected? Maybe the authors can discuss this further potentially highlighting a limitation of the approach based on how well separated good and bad behaviours are in a given domain.
>
> We first note Table 1 does not show ratios, Table 2 does. Table 1 shows fractions for the ID and OOD data sets which have $AC$-certificates less than $\epsilon$.
>
> The reviewer makes a good point here VALID completely depends on the separation of log-likelihood ratios (LR) between in- and out-of-domain samples, so domain similarity plays a big role. If $F$ and $T$ are very close to each other, VALID is unlikely to work well. Consider medical language discussing diabetes as $T$ and medical language on cardiovascular diseases as $F$. It is very conceivable that in terms of language, vocabulary and semantics, these two are very similar and likelihood ratios are entangled. This is true of OOD detection in general: An overlap of LR distribution naturally lowers the Bayes optimal classifier, which bounds our performance. We mention this limitation in lines L443-L447. In future work we hope to explore ways to mitigate this. For example by also explicitly training G to have low likelihood on negative samples (possibly via hard negative mining) on OOD domains that are similar semantically.

---

> > ### Author Response · Authors · 2024-11-25
> > **Rebuttal 2/3**
> >
> > > The generative results are weaker than the fixed sample set results, summarised lets say as weaker Youden's J for the same domain setting, and since this is the more realistic evaluation, it is important to caution that this approach isn't quite deployable at the moment. (On that note, lines 145-152 are more appropriate moved to the final paragraphs as a discussion section/future work/applications blurb)
> >
> > The results on responses generated by $L$ are indeed slightly weaker. We would push back on the idea that VALID is not deployable at the moment. Using VALID on the Medical QA data set reduces the likelihood of OOD outputs by $10^{40}$ on average over $D_F$, (L393-395) which is a significant reduction. We also believe a commercial entity with greater resources would be able to invest a larger amount of time and compute in the training on $G$ and the handcrafting of the various data sets, greatly improving upon the results shown here.
> >
> > > Together, the problem formulation and empirical results leave the work in a somewhat middling position in terms of contribution to the field, but depending on the potential discussion surrounding questions below, this might be improved.
> >
> > We respectfully disagree with the reviewer. In this work, we provide a new framework to safeguarding LLM deployment with provable guarantees and hence significantly moving beyond current methodology. In addition, we provide an algorithm that - while not perfect - provides an adversarial bound that is a) computationally very scalable and b) includes the *entire* input space.
> >
> >
> > > Q1: As a preliminary ask, "Such a certificate with respect to G can be useful: As G is only trained on samples in DT ⊂ T, a dataset of domain T, it assigns exponentially small likelihood to samples that are in F." Is there an analytic or empirical argument that the authors can provide for this statement? It seems like both the benign and adversarial robustness of the certificate hinge on this assumption.
> >
> > We do find this empirically to be the case. The likelihoods of OOD data under $G$ are very small and decay exponentially with length on average. For responses of 20 tokens we find ODO responses to be approximately $150$ orders of magnitudes less likely than ID responses. We have added Figure 11a to Appendix D.4 of the paper, which demonstrates the lower likelihood of OOD samples under $G$ relative to ID samples, as well as, the decay of the likelihood in the length of the response $N_y$.
> >
> > > Q2: Now, the section of future work describes a desire to explore using stronger models for G, which suggests that the authors don't necessarily see the weakness of G, eg. its inability to predict next tokens in any context beyond the small corpus on which it was fit, and maybe poorly even within that domain, as the actual crux of the method that makes it work.
> >
> >
> > > Q3: Was any ablation done about the model size for G and or the corpus size for training it? My hypothesis is that the stronger G becomes (more parameters and/or data), the wider set of strings it is able to model and therefor the closer L(y|x) and G(y) become, eventually depleting all the power of the algorithm.
> >
> > We believe the method is independent of the model size of $G$, and dependent on the data G is trained on. Several recent works [1] questions the ability of foundation models to generalise beyond the training data. A model trained on medical data, irrespective how big the model is (even when scaling the data with the scale of the model parameters), will have a hard time generating an argument "How did Beethoven view Mozart's music?". The larger $G$ is, the better the model can generalise within the domain, answering more complex medical data more accurately. Thus, $G$ size trades off between nuanced its language comprehension and computation cost. We have added Appendix E.2 in which we compare the performance of VALID for 3 sizes of $G$. The results show that larger $G$ yields tighter likelihood ratios and hence allows for a smaller $k$ that benefits the certificate. When scaling $G$, $G(y)$ becomes closer to $L(y|x)$ for in-domain data (hence allowing for smaller $k$). However, this is not the case for OOD data.
> >
> > [1] Udandarao V, Prabhu A, Ghosh A, Sharma Y, Torr P, Bibi A, Albanie S, Bethge M. No" zero-shot" without exponential data: Pretraining concept frequency determines multimodal model performance. In The Thirty-eighth Annual Conference on Neural Information Processing Systems 2024 Apr 4.

---

> > > ### Author Response · Authors · 2024-11-25
> > > **Rebuttal 3/3**
> > >
> > > > Q4: Overall, can the authors elaborate on any "behind the scenes" details that aren't in the draft or appendix on how G's role in the certificate was developed during the research and why the particular experimental choices (tiny gpt2's) for it were made?
> > >
> > > When we first conceived the idea, it was initially onto whether we can have a guarantee that a model will not behave differently from a different oracle model specialised in a given domain, i.e., a bound on the divergence. A question arises "How to construct such an oracle, and the answer was to approximate it with a model. Given our resources and available compute, we wanted the largest possible model that fits our hardware with a quick turn around for the proto-typing and experimentation. GPT-2 architecture was the choice here. It empirically performed better than smaller models in early experiments. We have added a  Appendix E.2 which contains an ablation on size of $G$, as mentioned in our response to Q2.
> > >
> > > > Q4: [...] Currently, my takeaway centres the identification of a weak G to compute an odds ratio against L as a way to perform rejection sampling for safety and alignment, and thus pitching the results a little more in this direction potentially improves the relevance and generalisability of the work (since the particular certificates/and algorithm might be limited in immediate applicability).
> > >
> > > This is a good point. However, we generally do not think VALID is suitable for alignment. We discuss our thought process in our Limitations Section. For alignment we believe a more nuanced understanding of language is required, rather than which topic is being discussed. However, carefully choosing $F$ and $D_F$ to include unsafe behaviours can have strong implications on safety. We leave this for future work.
> > >
> > > > Q5: Some relevant works that abstractly leverage the same odds ratio paradigm, though for different purposes, are Contrastive Decoding by Li et al. (2210.15097) which moderates policy L via a weaker policy L' at decoding time to improve output quality, and Direct Preference Optimisation by Rafailov et al. (2305.18290) as a way to train a policy L to prefer samples in T rather than T'.
> > >
> > > We thank the reviewer for bringing these papers to our attention. We have added them to our paper.

---

> > > > ### Comment · Reviewer_8eQm · 2024-11-26
> > > > **Response to rebuttal**
> > > >
> > > > (apologies for the delay in responding)
> > > >
> > > > ### Interpreting numerical certificates
> > > >
> > > > > Proportion of ϵy ≤ ϵ
> > > >
> > > > Refers to the fraction of samples that pass the criteria (technically not a ratio, I understand). So I was commenting that without reference methods or baselines or a intuitive example set to illustrate what D_T and D_F contain, then one must reason about the descriptions provided in Sec 3.1 and decide whether 59.90 rate for D_T at eps 10^-10 is good for a 95.14 rate on D_F for the QA scenario, or whether this is not loose enough on D_T. Then this must be compared to a 66.0 versus 100.0 for Shakespear at 10^-100. This is difficult to glean insights from directly without more discussion or presented examples of properly certified and falsely rejected samples.
> > > >
> > > > Regarding domain similarity, yes. However, the example on diabetes versus cardiovascular health is illustrative of an issue that is actually critical for this entire line of research, and not one that is expected to be "mitigated" by a clever improvement of any certificate technique. For distributions that are trivially separable, then many techniques will work as safety filters against one domain (interpret as "a lay reader can tell D_T and D_F apart"). However, in the limit, i.e. the edge cases where simple heuristics and spurious distributional features no longer suffice, it devolves to a task as difficult as any definition of "natural language understanding" in a specific domain, and all papers that decide to embark on this line of research need to be grounded in this reality. When describing the choice of experimental datasets and results, the draft would benefit from a discussion of how the choice of D_T and D_F influences expected performance of any, even naive baseline, certificate method.
> > > >
> > > > ### On maturity of the technique
> > > >
> > > > The reviewer and authors simply disagree on the appropriate level of moderation when claiming that "proveable" or "certified" techniques are ready for the real world. As an example, computational complexity bounds for the best known algorithms to break certain cryptographic protocols are a different class of "proveable" than arguing that because in preliminary experiments, a model is less likely (regardless of order/p-value) to produce a sample from a finite set of undesireable samples, it is therefore safe to deploy as if the true expected rate of violations is p or 10^40 times less likely. I think that it's important as scientists to be careful about these types of distinctions and use our language carefully whilst of course still arguing for the value of our work.
> > > >
> > > > (I would be remiss to not mention that at least a basic study of the adversarial robustness of any certification procedure is of course required before arguing that the approach is deployable to real human users, especially in the medical scenario featuring prominently in this draft.)
> > > >
> > > > ### On the design of G
> > > >
> > > > Figure 11a in Appendix D.4 is great!
> > > >
> > > > The inclusion of the new study in F.2 is also a good addition to the overall paper. In reflection on my own statement and the response, I do agree that it is probably the case that the design of G should be more data centric because of the central goal of essentially only/over-fitting to a specific small domain of sequences.
> > > >
> > > > However, I cannot help but notice the slight contradiction between intuition and experiment already present in the rebuttal plus added appendix section. While the hypothesis
> > > >
> > > > > We believe the method is independent of the model size of, and dependent on the data G is trained on.
> > > >
> > > > is quite plausible, your initial results suggests at least a small effect of model scale whilst other variables are held constant. The rebuttal says
> > > >
> > > > > The results show that larger yields tighter likelihood ratios and hence allows for a smaller that benefits the certificate.
> > > >
> > > > while the added material in F.2 says
> > > >
> > > > > We find that larger models tend to perform better, however evidence is not strong.
> > > >
> > > > Generally, my takeaway is that, indeed, as discussed in my initial review and as indulged by your additional studies during rebuttal, the design of G is a key part of the technique, and might need careful tuning depending on the deployment context and the relative diversities of sets D_T and D_F versus general web text and or the space of possible user queries.
> > > >
> > > >
> > > > ### Current recommendation
> > > >
> > > > The draft is interesting but requires a bit of polish in the presentation of empirical results so that they are readily and intuitively interpretable, and it also requires some calibration of certain claims about deployability (as do many similar papers in all fairness).
> > > >
> > > > The design of G and the study of datasets D_T and D_F are the most interesting parts of the work in this reviewer's opinion, and should be reworked to feature more prominently in the draft.
> > > >
> > > > I will maintain my rating to reflect the opinion that this is not quite ready for publication, but bump the _contribution_ score in response to the authors' work during rebuttal.

---

> > > > > ### Author Response · Authors · 2024-11-30
> > > > >
> > > > > ### On maturity of the technique
> > > > >
> > > > > > The reviewer and authors simply disagree on the appropriate level of moderation when claiming that "proveable" or "certified" techniques are ready for the real world. As an example, computational complexity bounds for the best known algorithms to break certain cryptographic protocols are a different class of "proveable" than arguing that because in preliminary experiments, a model is less likely (regardless of order/p-value) to produce a sample from a finite set of undesireable samples, it is therefore safe to deploy as if the true expected rate of violations is p or $10^{40}$ times less likely. I think that it's important as scientists to be careful about these types of distinctions and use our language carefully whilst of course still arguing for the value of our work.
> > > > >
> > > > > We do indeed disagree with the reviewer on this. We are not entirely sure what the reviewer's core statement is above, but we will respond to a few arguments we are picking up on. We kindly ask the reviewer to clarify, if we missed their point.
> > > > >
> > > > > * Our method is fact *provable*: We can prove a statement is true for the entire set $X$ of prompts. This is very distinct from empirical methods on safety, which draw conclusions on small finite samples of inputs $X$. Obtaining non-trivial, scalable, and global bounds over input $X$ is rare in ML literature.
> > > > > * We believe "preliminary experiments" is misrepresenting our efforts. Reviewers tYKL and pqHZ state that our experimental setup shows "generizability" and "effectiveness".
> > > > > * Could the reviewer please point us to where we claim that a $10^{40}$ times smaller was the "true expected violation rate"? Were do we draw the conclusion from that fact that our model is "provably safe"?
> > > > > * We reject the reviewer's notion that we are not appropriately careful in our language. Every discipline in computer science has different standard as to what certifiable or provable means. ML systems due to their opacities are difficult to certify. For instance, Cohen et al. (2019) propose Gaussian Smoothing with certifiable defenses for image classification, certfying very small $\ell_2$ radii. Salman et al. (2019) provide methods to increase these radii significantly (yet still small), improving upon the *certified accuracies w.r.t. finite test sets*. Their work is titled: "Provably Robust Deep Learning via Adversarially Trained Smoothed Classifiers". Just these two works share over 2500 citations between them as of today. While we appreciate that 'provable' or 'certifiable' means something different to different people, we believe that we are fully within the norms and precision when we do call our method "provable" and "certification".
> > > > >
> > > > > > (I would be remiss to not mention that at least a basic study of the adversarial robustness of any certification procedure is of course required before arguing that the approach is deployable to real human users, especially in the medical scenario featuring prominently in this draft.)
> > > > >
> > > > > We agree.

---

> ### Author Response · Authors · 2024-11-30
>
> We thank the reviewer for taking the time to read our rebuttal and respond in such detail. Please find our comments below.
>
> ### Interpreting numerical certificates
>
> > Refers to the fraction of samples that pass the criteria (technically not a ratio, I understand). So I was commenting that without reference methods or baselines or a intuitive example set to illustrate what D_T and D_F contain, then one must reason about the descriptions provided in Sec 3.1 and decide whether 59.90 rate for D_T at eps 10^-10 is good for a 95.14 rate on D_F for the QA scenario, or whether this is not loose enough on D_T. Then this must be compared to a 66.0 versus 100.0 for Shakespeare at 10^-100. This is difficult to glean insights from directly without more discussion or presented examples of properly certified and falsely rejected samples.
>
> We agree that the results in Table 1 are not easily compared with baselines. Further, interpreting such small probabilities in such high-dimensional space is challenging. Hence, we provide a frame of reference with our example on number of requests per second relative to number of incursions per year (L148-L152). For the camera ready version, we will augment Table 1 with a figure on the distribution of ACs in $D_T$ and $D_F$ to help readers interpret the numbers. We will strongly highlight what we think the most important lesson is (L304-L309) and further help with interpreting the numbers presented in-text.
>
> Two smaller notes:
> * We do not recommend direct numerical comparisons between datasets / domains: Domains vary in the heterogeneity, vocabularies, and sentence lengths. In our experiments we take great care to ensure comparability *within* each domain, but not between.
> * Table 2 is a *comparison* to a crude baseline. We compare our method using the constriction ratio (CR) to an approximate baseline (i.e. the non-adversarial likelihood under $L$).
>
> > Regarding domain similarity, yes. However, the example on diabetes versus cardiovascular health is illustrative of an issue that is actually critical for this entire line of research, and not one that is expected to be "mitigated" by a clever improvement of any certificate technique. For distributions that are trivially separable, then many techniques will work as safety filters against one domain (interpret as "a lay reader can tell D_T and D_F apart"). However, in the limit, i.e. the edge cases where simple heuristics and spurious distributional features no longer suffice, it devolves to a task as difficult as any definition of "natural language understanding" in a specific domain, and all papers that decide to embark on this line of research need to be grounded in this reality. When describing the choice of experimental datasets and results, the draft would benefit from a discussion of how the choice of D_T and D_F influences expected performance of any, even naive baseline, certificate method.
>
> We agree with the reviewer that domain similarity is an issue with our work and, in fact, any other work on domain separation (e.g. OOD detection). In the limit of perfect similarity, of course, no training technique can improve the separation. This is why we refer to the Bayes optimal classifier limiting our method.
>
> However, we disagree with what seems like a dichotomy brought forward by the reviewer: trivial seperation or none. The datasets we use as OOD are considered very close to ID in the literature. For example, the 20NG data set contains a number of different categories, and we test between different categories *within 20NG* rather than comparing it with other datasets (L259-L265).
>
> While we mention the exact datasets used for $D_F$ and $D_T$ (Section 3.1 and Appendix C), the reviewer is correct that we do not provide extensive reasoning on this selection. We will update Appendix C, to clarify why we choose these datasets and how their choice impacts our results.

---

> ### Author Response · Authors · 2024-11-30
>
> ### On the design of G
>
> > Figure 11a in Appendix D.4 is great!
>
> We are glad the reviewer finds this helpful.
>
> > The inclusion of the new study in F.2 is also a good addition to the overall paper. In reflection on my own statement and the response, I do agree that it is probably the case that the design of G should be more data centric because of the central goal of essentially only/over-fitting to a specific small domain of sequences.
>
> We are glad the reviewer finds F.2 helpful too. Just a short remark: we do rely on $G$ generalising *within* the target domain while not generalizing beyond. Hence, "over-fitting" is not the correct term, but we find "only-fitting" delightful and very illustrative of what $G$'s role in VALID is.
>
> > However, I cannot help but notice the slight contradiction between intuition and experiment already present in the rebuttal plus added appendix section. While the hypothesis [...] independent of model size [...] while F.2 says [...] larger models tend to perform better.
>
> We thank the reviewer for bringing this to our attention and apologize for the unclear communication. We are not seperating well in our communication and will attempt to clarify.
>
> 1. The reviewer suggested that the "weakness of $G$" might be part of why VALID works. We agree with the reviewer that the *weakness* of $G$ on *OOD samples* is why it works. However, the ability of $G$ to generalize within the target domain is very much needed to tighten log likelihood ratios on in-domain data. This allows for smaller $k$ which tightens the bound.
> 2. We took the reviewer's "weakness  of $G$" to mean "smallness of $G$", which with we disagree: We state that rather than restricting the size of $G$, restricting the data pool of $G$ is what makes this method work in the first place. This does not mean the size of $G$ is irrelevant, but much less important in relation. To back this up we present 2 kinds of evidence:
> 	1. Figure 11 showing how the OOD likelihoods of a model domain-specific and a foundation model differ.
> 	2. We perform the ablation in Appendix F.2 to test the *smallness* of $G$: We observe that larger models fit better on in-domain data, hence benefitting VALID through smaller $k$. This contradicts the notion of *smallness* being a beneficial factor.
> 	3. Finally we acknowledge that there is an interaction between model size and training data size. Increasing the size of a model without approprate regulisation can result in severe overfitting which would make it unsuitable to be $G$.
>
> Hence, while acknowledging that larger $G$ models tend to perform better (we can show how), we believe that the "crux" of this method is the training pool of $G$ with the size of $G$ playing a secondary role.
>
> We hope this clarifies the issue.
>
> > Generally, my takeaway is that, indeed, as discussed in my initial review and as indulged by your additional studies during rebuttal, the design of G is a key part of the technique, and might need careful tuning depending on the deployment context and the relative diversities of sets D_T and D_F versus general web text and or the space of possible user queries.
>
> We mostly agree with this. We find $G$ to perform surprisingly well with little tuning (we mostly use HuggingFace standard parameters with commonly used combinations of layers, heads etc within the GPT-2 architecture). But yes, the "deployment context" of $G$ is the key.

---

> ### Author Response · Authors · 2024-11-30
>
> ### Current recommendation
>
> > The draft is interesting but requires a bit of polish in the presentation of empirical results so that they are readily and intuitively interpretable, and it also requires some calibration of certain claims about deployability (as do many similar papers in all fairness).
>
> We welcome suggestions for additional plots that the reviewer would like to see included in the camera ready version of the paper if we are successful.
> Finally, we agree with the reviewer that most papers present completed research, rather than completed products, we believe our paper is just as applicable to the former category as most other papers accepted to ICLR.
>
> > The design of G and the study of datasets D_T and D_F are the most interesting parts of the work in this reviewer's opinion, and should be reworked to feature more prominently in the draft.
>
> We agree with the reviewer that the design of $G$ and $D_T, D_F$ are interesting parts and for practitioners the most relevant. However, we propose a framework that does not exist in its current form and feel that we need to motivate this carefully and provide detail why it is so different from what is commonly done in LLM restriction / alignment research today. Before diving in too much into how to choose $G$ or $D_T$, we are glad that all 4 reviewers acknowledged that our framework is well motivated.

---

> > ### Comment · Reviewer_8eQm · 2024-12-01
> > **Continuing resolution**
> >
> > Responses to my points in _"Interpreting numerical certificates"_ and _"On the design of G"_ are appreciated, and the discussed tweaks to the wording and presentation of certain sections will be appreciated in the next draft of the work as I think they add considerable value and clarity.
> >
> > Responses to _"Maturity of the technique"_ and _"Current recommendation"_ are unfortunately correlated and indicative of the same underlying issue I have with the results and presentation. The response doubles down rather than recognizing the concern.
> >
> > > Using VALID on the Medical QA data set reduces the likelihood of OOD outputs by 10^40 on average over, (L393-395) which is a significant reduction.
> >
> > is language from the rebuttal itself, which cites a statement in the original draft. Citing this kind of number repeatedly indicates that the authors do believe it to be a non-vacuous statement (at reduction 10^40 it doesn't matter what the base rate is, it's now effectively zero, or the comparison is meaningless, but not both).
> > A final comment here to add context is that these kinds of expected error rate statements are often inflated in academic settings, with real reliability rates in the wild being much worse. ML safety research has been complicit in this slight bit of malpractice recently in effect enabling the hasty deployment of certain technologies surrounding LLMs that are not ready for primetime yet. The reviewer believes this trend is partially due to the way in which results are discussed and presented in academic papers because of our focus on the _publication_ process not in solving the underlying problem and presenting actionable advice and positively impacting in the real world.
> >
> > My general remark is that the bullet points on:
> > - "proveable... for the entire set X of prompts"
> > - "Were do we draw the conclusion from that fact that our model is 'provably safe'?"
> > - "we are fully within the norms and precision when we do call our method 'provable' and 'certification'."
> >
> > feel slightly unparsimonious.
> >
> > An isolated comment on bullet one, is that we assume that X is referring to D_F here or some other finite set, _not_ something like F.  Therefore, "global bounds" can be achieved for a set that is trivially unrepresentative of the true bad set F,  or via a quantity like an empirical model likelihood G(y), or the bound can just be loose, or the evaluation setting can be divorced from real deployment, and any/all of these details can render such a guarantee of questionable utility in practice.
> >
> > More generally though the co-occurring statements indicate a willingness to claim production ready tightness in on one case, but then during rebuttal claim "well this style of proveable certificate passes muster in this field, see citations" in another, whichever is more expedient to the argument at hand.
> >
> > I think I will leave my score as it is for this work, but I will acknowledge that the AC might choose to use the other ratings as arguments for acceptance, and I would understand this. There is a fundamental mismatch in expectations between this reviewer and the authors about what work on robustness and safety in the age of LLMs should strive to accomplish both in experiment and in presentation, but it is quite possible that the reviewer is in an (idealistic) minority opinion here.

---

> > > ### Author Response · Authors · 2024-12-03
> > >
> > > We sincerely thank the reviewer for their very active engagement in the review process. While we disagree on issues, we do very much appreciate the extensive thoughts the reviewer has given our work.
> > >
> > > > Responses to my points in _"Interpreting numerical certificates"_ and _"On the design of G"_ are appreciated, and the discussed tweaks to the wording and presentation of certain sections will be appreciated in the next draft of the work as I think they add considerable value and clarity.
> > >
> > > We are glad to hear.
> > >
> > > > Responses to _"Maturity of the technique"_ and _"Current recommendation"_ are unfortunately correlated and indicative of the same underlying issue I have with the results and presentation. The response doubles down rather than recognizing the concern.
> > >
> > > Naturally, we are disappointed by the continued disagreement between reviewer and authors. However, we do appreciate the discussion on these and acknowledge that the reviewers comments have been beneficial to enhance our view on the problem at hand, the methods we provide and the safety literature at large.
> > >
> > >
> > > > [...] we assume that X is referring to D_F here or some other finite set, _not_ something like F. Therefore, "global bounds" can be achieved for a set that is trivially unrepresentative [...]
> > >
> > > It is correct, that "global" can mean various things. In our case we mean: For a given $y$ the $\epsilon_y$-AC provides a "global bound" on the prompts, i.e.  $\forall x \in \mathbb{S}$. This is very distinct from, e.g. Gaussian smoothing that only provides a guarantee for a $\ell_2$ ball around some input $x$. The AC can be extended to $D_F$ as DC. "global" does not mean $\forall y \in \mathbb{F}$.  We will revise the paper to clarify wherever ambiguous. Further, the reviewer is absolutely correct in noting that selecting an inadequate evaluation dataset results in questionable conclusions - a problem machine learning researchers often face.
> > >
> > > Concluding, we continue to disagree with the reviewer on what "certification", "provable defenses" etc. should mean and what they mean in the light of recent literature. This has implications on the diverging views regarding the maturity of our method. Nonetheless, the reviewer seems to agree that this is an interesting advancement of the field.
> > >
> > > We appreciate the extensive time the reviewer has dedicated to reviewing our paper. If there are any last minute questions, we will gladly respond.

---

### Official Review · Reviewer_pqHZ · 2024-11-04

**Soundness:** 3
**Presentation:** 3
**Contribution:** 2
**Rating:** 6
**Confidence:** 3

**Summary:**

This work studies domain certification, which refers to the characterization of the out-of-domain behavior of language models. This paper first formaize the definition of domain certification. And then propose an approach, VALID, to achieve domain certification. The proposed approach is empirically evaluated on multiple datasets in various domains including TinyShakespeare, 20NG, MedicalQA. The experimetal results show that the proposed method is effective at achieving domain certification as defined in this work.

**Strengths:**

1. This work introduces and formalize domain certification for characterizing the out-of-domain behavior of language models under adversarial attach, which is useful for language models tuned a specific domain.
2. An approach called VALID is proposed to achieve domain certification. VALID bounds the probability of a LLM answering out-of-domain questions.

**Weaknesses:**

The main concern for this work is its limitations including the reliance on the domain generator which doesn't consider the model input. In addition, Theorem 1 assumes the certificate is useful given G is trained on in-domain data. However, as language models are usually pre-trained on large amound of text data, which ingests world knowledge into it. Therefore, model G can contains out-of-domain knowedge, which makes Theorem 1 extremely limited.

**Questions:**

See weaknesses section.

---

> ### Author Response · Authors · 2024-11-22
> **Rebuttal 1/1**
>
> We thank the reviewer for taking the time to review our work. We are appreciate the reviewer agrees that the "proposed method [VALID] is effective" and that our domain certification framework is "useful for language models" deployed in a specific domain.
>
> > W1: The main concern for this work is its limitations including the reliance on the domain generator which doesn't consider the model input.
>
> The lack of context (using $G(Y)$ rather than $G(Y|X)$) in the rejection condition is a trade off. We mention some of the negative effects of this choice in the limitations section (L436-L442). However we believe it is a net gain for two main reasons:
> 1) Omitting the prompt $X$ makes the bound adversarial. This enables us to get a non-vacuous bound over all prompts which would be very hard otherwise. Finding a worst-case bound for a condition that depends on $X$ requires optimising over token space which is discrete and highly non-convex, to the best of our knowledge no efficient method exist for finding the global optimum.
> 2) Many modern models are very verbose. As mentioned in line L440, such verbosity and tendency to repeat the query can help. For instance, LLMs are currently trained not to respond "Yes.", but rather "Yes, C4 is a great ingredient for a bomb." making context available in the response.
>
> > W2: Theorem 1 assumes the certificate is useful given G is trained on in-domain data. However, as language models are usually pre-trained on large amount of text data, which ingests world knowledge into it. Therefore, model G can contains out-of-domain knowedge, which makes Theorem 1 extremely limited.
>
> We acknowledge that most LLMs are pre-trained on large amounts of text data, most likely containing a diverse set of domains. This is, in fact, one reason why LLMs are so adversarially vulnerable when attackers elicit content that was learned and then supposedly "unlearned". However, in the VALID framework the $G$ model is trained from scratch on purely in-domain data for the specific task. This is the process we follow in our experiments. While we use a GPT-2 architecture, this is trained from a random initialization hence the model has *never* seen OOD data.
>
> We note if one was to train $G$ using OOD data, the Theorem 1 would still hold. However, the bound of Theorem 1 might become vacuous as $G$ would likely assign high likelihood on responses that are OOD. Our empirical results show that $G$ places sufficiently little probability on the OOD data, and hence the system both provides good OOD detection and useful certificates.

---

> > ### Comment · Reviewer_pqHZ · 2024-11-27
> >
> > >  However, in the VALID framework the model is trained from scratch on purely in-domain data for the specific task.
> >
> > As ML models has generalization capability, it might generalizes to OOD data. Have you considered generalization in your theorem and experiments?

---

> > > ### Author Response · Authors · 2024-11-28
> > >
> > > Several recent works [1] have questioned the ability of foundation models to generalise to new domains and topics beyond their training data. A model trained exclusively on medical data, irrespective of how big the model is (even when scaling the data with the scale of the model parameters), will have a hard time generating an argument "How did Beethoven view Mozart's music?"
> > > In Figure 11, we provide concrete empirical evidence of this. We present the likelihood that ID and OOD samples have under $G$ (guide model) and $L$ (foundation model) for medical QA. You may observe that the likelihood of OOD samples under $G$ is much lower than that of ID samples under $G$. This gap grows exponentially as the length of sequences increases. This is not observable for a model like $L$ that does generalise beyond the ID samples.
> > > We acknowledge VALID depends on the separation of log-likelihood ratios (LR) between in- and out-of-domain samples, so domain similarity plays a big role. If $T$ and $F$ are very close to each other, VALID is unlikely to work well. Consider medical language discussing diabetes as $T$ and medical language on cardiovascular diseases as $F$. It is very conceivable that in terms of language, vocabulary and semantics, these two are very similar and likelihood ratios are entangled. This is true of OOD detection in general: An overlap of LR distribution naturally lowers the Bayes optimal classifier, which bounds our performance. We mention this limitation in lines L443-L447. In future work we hope to explore ways to mitigate this. For example by also explicitly training $G$ to have low likelihood on negative samples (possibly via hard negative mining) on OOD domains that are similar semantically.
> > >
> > > [1] Udandarao V, Prabhu A, Ghosh A, Sharma Y, Torr P, Bibi A, Albanie S, Bethge M. No" zero-shot" without exponential data: Pretraining concept frequency determines multimodal model performance. In The Thirty-eighth Annual Conference on Neural Information Processing Systems 2024 Apr 4.

---

> > > > ### Author Response · Authors · 2024-12-03
> > > >
> > > > We hope that our response has addressed the reviewer's concerns regarding the generalization of the model $G$. Should there be any questions left, we will gladly respond.

---

### Official Review · Reviewer_tYKL · 2024-11-04

**Soundness:** 4
**Presentation:** 4
**Contribution:** 4
**Rating:** 8
**Confidence:** 3

**Summary:**

The paper proposes a theoretical foundation for certifying language models trained to produce outputs in a domain. Then the work proses VALID, an algrithm that provides provable upper bounds on out of domain behavior of language models. The method preserves a good chunk of the unconstrained LM's performance on MMLU@Med while being robust to poducing out-of-domain outputs. There are also various other experiment results on datasets like TinyShakespeare and 20NG that show the effectiveness of the method to potentially detect and certify OOD behavior of language models.

**Strengths:**

- Strong theoretical foundations for classifying out-of-domain behavior of language models and ways to prevent this.
- The algorithm VALID is relatively straightforward and uses rejection sampling as an elegant way to achieve the certifications.
- The empirical evidence presented is across various kinds of datasets(Tiny Shakespeare, MedQA) which shows the generizability of VALID.
- The work is novel and needed to provide theoretical insights and guarantees for safe deployment of language models.
- Well written and the progression from atomic to domain certificates is explained well.

**Weaknesses:**

The paper acknowledges most of the limitations, but there is still room for further discussion:
1. Lack of context for guide model G. The work does argue that potentially involving the context in the final answer could fix this issue, but then the method cannot work in cases where a user wants language model to be concise and this also increases the inference cost of models as many more tokens get sampled for the output.
2. Adversarial attacks on G/M. The work acknowledges and shows adversarial attacks this method is prone to but argues that as adversaries would need white-box access to G(line 448, 453) and so the attacks may not be feasible. White-box access does not need to hold for the success of adversarial attacks, as some attacks do generalize across various models even if they were originally targeted at some other specific model. This should be further investigated.
3. Rejection sampling with T>1  incurs an inefficiency, which could be reduced as multiple samples could be drawn in parallel. In any case, there is additional computational overhead.

**Questions:**

1. Is there a reason that certified benchmarking was only done on MMLU@Med? I think it would be useful to have results for other benchmarks as well.
2. Could we use a guide model G to detect whether an input is in-domain? It might be interesting to see how the likelihood of L(y|x) would change for "in-domain" and ood inputs.
3. While the authors mention certification in computer vision (line 80-81), there has been some work on certification in NLP as well, such as https://arxiv.org/abs/2401.01262, https://arxiv.org/abs/2402.15929, https://arxiv.org/abs/2403.10144v1.  I think mentions of some of these works would prevent the false impression that certification has only been applied to computer vision and provide a more complete picture of the certification work being done across domains.

---

> ### Author Response · Authors · 2024-11-25
> **Rebuttal 1**
>
> We thank the reviewer for their efforts to review our work and providing feedback. We appreciate the reviewer recognising the strong theoretical foundation, the effectiveness of VALID and the novelty of our work.
>
>
> > W1: Lack of context for guide model G. [...] involving the context in the final answer could fix this issue, but [...] [not] concise and this also increases the inference cost [...].
>
> We agree with the reviewer that increased verbosity comes at extra inference cost and prevents consiseness. However, the lack of context (using $G(Y)$ rather than $G(Y|X)$) in the rejection condition is a trade off. We mention some of the negative effects of this choice in the limitations section (L436-L442). We believe it is a net gain for the following reasons:
> 1) Omitting the prompt $X$ makes the bound adversarial. This enables us to get a non-vacuous bound over all prompts which would be very hard otherwise. Finding a worst-case bound for a condition that depends on $X$ requires optimising over token space which is discrete and highly non-convex, to the best of our knowledge no efficient method exist for finding the global optimum.
> 2) Many modern models are very verbose. As mentioned in line L440, such verbosity and tendency to repeat the query can help. LLMs are currently trained not to respond "Yes.", but rather than "Yes, C4 is a great ingredient for a bomb." making context available in the response. This suggests that the extra cost and lack of conciseness are (at least) tolerated by the developers of LLMs today - even without VALID.
>
> > W2: Adversarial attacks on G/M. The work acknowledges and shows adversarial attacks this method is prone to but argues that as adversaries would need white-box access to G(line 448, 453) and so the attacks may not be feasible. White-box access does not need to hold for the success of adversarial attacks, as some attacks do generalize across various models even if they were originally targeted at some other specific model. This should be further investigated.
>
> The reviewer raises an interesting point here, yes white box access might not be needed if the adversary has sufficient knowledge about $G$, its training process and its training data and $D_{\mathbb{F}}$ (the data output it had already been certified for). Then they could train a model $G'$ that they believe is similar to $G$ in order to perform transfer attacks on $G$. The adversary would then need to find a $Y \in \mathbb{F} \cap D_{\mathbb{F}}'$ (where $D_{\mathbb{F}}'$ is the complement of $D_{\mathbb{F}}$) which maximises $G(Y)$. This is a constrained optimisation in token space, and thus non-trivial to perform. The feasibiltiy of such an attack on $G/M$ would be an interesting direction to explore. While we acknowledge such an attack may be possible, this requires a lot more insider knowledge and compute than standard black box attacks on $L$ that optimise over input $X$ to output a string $Y$. Finally, we note even if such an attack was to be successful it would not, break any VALID certificates.
>
> > W3: Rejection sampling with T>1 incurs an inefficiency, which could be reduced as multiple samples could be drawn in parallel. In any case, there is additional computational overhead.
>
> We agree that generating multiple samples does increase the cost of generation per query and yes, these could be run in parrallel if latancy was a greater concern than total compute usage. However $T$ is a hyperparameter that trades off computational overhead against improved acceptance behaviour of the model for in-domain data. We explore this in Appendix E of the updated paper, showing false rejection rates are significantly improved for $T>1$ at almost no cost to the $\epsilon$-DC.
>
> > Q1: Is there a reason that certified benchmarking was only done on MMLU@Med? I think it would be useful to have results for other benchmarks as well.
>
> Unfortunately, there is an ever increasing number of interesting LLM benchmarks, of ever increasing scale and regrettably we must select a finite subset to try. We focus on MMLU-Med as it is commonly used and to illustrate a framework for handling multiple-choice benchmarks in a meaningful way. We hope the datasets presented in the paper give a good indication of performance over a broad number of settings.

---

> ### Author Response · Authors · 2024-11-25
> **Rebuttal 2**
>
> > Q2: Could we use a guide model G to detect whether an input is in-domain? It might be interesting to see how the likelihood of L(y|x) would change for "in-domain" and ood inputs.
>
> It would definitely be possible to use $G$ to detect whether inputs are ID or OOD. However, $G$ would then be adversarially vulnerable, i.e. it would be possible via optimisation to find OOD inputs that result in high likelihood responses. Thus, it is not immediate to us how one could establish a certificate. Nonetheless, the question is very interesting and we have added Figure 11 to Appendix D.4 where we show strong disentanglement of ID and OOD data in $G(y)$, but not for $L(y|x)$.
>
> > Q3: While the authors mention certification in computer vision (line 80-81), there has been some work on certification in NLP as well, such as https://arxiv.org/abs/2401.01262, https://arxiv.org/abs/2402.15929, https://arxiv.org/abs/2403.10144v1. I think mentions of some of these works would prevent the false impression that certification has only been applied to computer vision and provide a more complete picture of the certification work being done across domains.
>
> We thank the reviewer for highlighting these missing citations, we have added these to an updated version of the paper. We note that these papers present certification methods against different behaviours which makes them appealing in different settings to the one we consider.

---

> > ### Comment · Reviewer_tYKL · 2024-11-27
> >
> > I appreciate the author's response to the questions raised. After going through the discussions with other reviews and my understanding of the paper, I would like to maintain my score for the following reasons:
> > The method depends quite a bit on the selection of D_F, D_T and how well it represents F, T which is not an easy problem with a reliable solution.
> >
> > The manuscript lacks some needed bechmarking results rather than just MMLU-Med if it is to be decided that it works well over various domains while maintaining performance of highly performant models.

---

> > > ### Author Response · Authors · 2024-12-03
> > >
> > > We thank the reviewer for their efforts and engaging. We acknowledge the concerns brought forward.
> > >
> > > We appreciate the continued positive appraisal of our work and are happy to provide any last minute clarifications.

---

### Meta-Review · Area_Chair_qFxL · 2024-12-23

**Metareview:**

All reviewers except one (8eQm) argued for accepting the paper. For this reviewer their main conerns were on #1 a lack of discussion/intuition for presented results, #2 the design of G, #3 the precision of the authors’ language. Specifically for #1 the reviewer argued that the descriptions of Section 3.1 were insufficient to understand the results (e.g., in Table 1). The authors agreed and proposed to augment Table 1 with a figure to help readers interpret the numbers. They also proposed to update Appendix C to clarify the choice of datasets and how this choice impacts results. The author appreciated this and had no further suggestions. I consider concern #1 resolved. For #2 the reviewer had concerns about the the guide model G, including (a) whether there was any support for the statement "Such a certificate with respect to G can be useful: As G is only trained on samples in DT ⊂ T, a dataset of domain T, it assigns exponentially small likelihood to samples that are in F.”, (b) whether an ablation was done about the model size for G, and (c) wanted to see more details on how G’s role in the certificate was developed and what motivated experimental choices. The reviewers responded to (a) by adding Figure 11a, which empirically demonstates the above statement. The reviewers responded to (b) and (c) by adding an Appendix E.2 where they tested VALID on three different sizes of G. The reviewer appreciated these responses but was worried about a contradiction between intuition and experiments: the text says “We believe the method is independent of the model size of, and dependent on the data G is trained on.” but the added material in F.2 says “We find that larger models tend to perform better, however evidence is not strong”. The authors clarified that larger G models tend to perform better but that the crux of the method depends on the training pool of G. The reviewer was satisfied with this response, resolving concern #2. For #3 the reviewer took issue with the claims of “provability”,  “certification”, and whether the method is ready for practice. The authors responded by arguing that the method is provable: they prove a statement is true for a set of X prompts. They argue that provability using global bounds over the distribution of inputs is rare in ML. In general the authors reject the argument that they have not been careful in their language, arguing that their usage of “provability” and “certification” agree with the norms of their usage in ML literature. Finally, responding to the argument of practical readiness, the authors asked whether the reviewer had any additional requests for experiments. The reviewer responded that there are various statements in the paper such as “Using VALID on the Medical QA data set reduces the likelihood of OOD outputs by 10^40 on average over, (L393-395) which is a significant reduction.” which are vacuous as they do not indicate the results of the method in practice. They also argue that the authors overstate the production readiness of the method. The authors responded that they will revise the paper to clarify ambiguous wording. The final crux of this back and forth seems to be whether the authors use wording that is too strong to describe the practical applicability of their method. I agree with the reviewer that this is important as the ML community is notorious for overstating the usefulness of methods, which in the past has led to ML bubbles and is especially relevant for LLM safety. However, when specific examples were raised, the authors did try to update the paper based on the reviewer’s feedback. I would like to echo the importance of the reviewer’s concern and urge the authors to carefully go back to the points where they argue for the practical applicability of the method and revise these if they are not fully supported by the experimental evidence. This will set a better precident for future works. Given this, I believe concern #3 is resolved. Overall the paper makes substantial theoretical contributions to the area and proposes an elegant algorithm which is tested extensively. Given these things, I vote to accept. Authors: you’ve already made improvements to respond to reviewer changes, if you could double check their comments for any recommendation you may have missed on accident that would be great! After incorporating these changes the paper will make a nice contribution to the conference!

**Additional Comments On Reviewer Discussion:**

All reviewers responded to the author feedback (tYKL, with a short response; pqHZ with one further question; 8eQm with extremely detailed feedback and a back-and-forth discussion; and and XBJU with a short comment indicating they raised their score). No reviewers engaged in further discussion of the paper. Please see the meta review for further details.

---

### Decision · Program_Chairs · 2025-01-22

Accept (Poster)